# Improving the Privacy and Practicality of Objective Perturbation for Differentially Private Linear Learners

**Rachel Redberg**
UC Santa Barbara

**Antti Koskela**
Nokia Bell Labs

**Yu-Xiang Wang**
UC Santa Barbara

## Abstract

In the arena of privacy-preserving machine learning, differentially private stochastic gradient descent (DP-SGD) has outstripped the objective perturbation mechanism in popularity and interest. Though unrivaled in versatility, DP-SGD requires a non-trivial privacy overhead (for privately tuning the model's hyperparameters) and a computational complexity which might be extravagant for simple models such as linear and logistic regression. This paper revamps the objective perturbation mechanism with tighter privacy analyses and new computational tools that boost it to perform competitively with DP-SGD on unconstrained convex generalized linear problems.

## 1 Introduction

The rise of deep neural networks has transformed the study of differentially private learning no less than any other area of machine learning. Differentially private stochastic gradient descent (DP-SGD) (Song et al., 2013; Bassily et al., 2014; Abadi et al., 2016) has thus gained widespread appeal as a versatile framework for privately training deep learning models.

How does DP-SGD fare on simpler models such as linear and logistic regression? The verdict is unclear. Clearly an algorithm capable of privately optimizing non-convex functions represented by millions of parameters is up to the computational task of fitting a linear model. A more pressing concern is that DP-SGD is up to *too* much. Look, for example, at the algorithm's computational complexity: DP-SGD requires $O(n^2)$ steps to achieve the optimal excess risk bounds for DP convex empirical risk minimization (Bassily et al., 2014).

DP-SGD furthermore takes after its non-private counterpart in sensitivity to hyperparameters. A poor choice of learning rate or batch size, for instance, could lead to suboptimal performance or slow convergence. There are well-established procedures for hyperparameter optimization that typically involve evaluating the performance of the model trained using different sets of candidate hyperparameters. But with privacy constraints, there is a catch: tuning hyperparameters requires multiple passes over the training dataset and thereby constitutes a privacy cost.

At best, existing work tends to circumvent this obstacle by optimistically assuming the availability of a public auxiliary dataset for hyperparameter tuning. More often the procedure for private hyperparameter selection is left largely to the reader's imagination. Only recently have Liu & Talwar (2019) and subsequently Papernot & Steinke (2022) studied how to obtain tighter privacy loss bounds for this task beyond standard composition theorems.

In the meantime, objective perturbation (Chaudhuri et al., 2011; Kifer et al., 2012) has been to some extent shelved as a historical curiosity. Sifting through the literature, we find that opinions are divided: some tout objective perturbation as "[o]ne of the most effective algorithms for differentially private learning and optimization" (Neel et al., 2020), whereas other works (Wang et al., 2017) dismiss

37th Conference on Neural Information Processing Systems (NeurIPS 2023).

objective perturbation as being impractical and restrictive. Some empirical evaluations (Yu et al., 2019; McKenna et al., 2021) suggest that DP-SGD often achieves better utility in practice than does objective perturbation; others (Iyengar et al., 2019) report the opposite.

Our goal in this paper is to lay some of this debate to rest and demonstrate that for generalized linear problems in particular, objective perturbation can outshine DP-SGD.

## 1.1 Our Contributions

- **We establish an improved $(\epsilon, \delta)$-DP bound for objective perturbation via privacy profiles, a modern tool for privacy accounting that bounds the hockey-stick divergence.** The formula can be computed numerically using only calls to Gaussian CDFs. We further obtain a *dominating pair* of distributions as defined by Zhu et al. (2022) which enables tight composition and amplification by subsampling of the privacy profiles.

- **We present a novel Rényi differential privacy (RDP) (Mironov, 2017) analysis of the objective perturbation mechanism. Using this analysis, we show empirically that objective perturbation performs competitively against DP-SGD with "honestly"** [1] **tuned hyperparameters.** The tightest analyses to date of private hyperparameter tuning are the RDP bounds derived in Papernot & Steinke (2022). This tool allows us to empirically evaluate objective perturbation against DP-SGD on a level playing field (Section 5).

- **We fix a decade-old oversight in the privacy analysis of objective perturbation.** Existing literature overlooks a nuanced argument in the privacy analysis of objective perturbation, which requires a careful treatment of the dependence between the noise vector and the private minimizer. Without assuming GLM structure, the privacy bound of objective perturbation is subject to a dimensional dependence that has gone unacknowledged in previous work[2].

- **We introduce computational tools that expand the applicability of objective perturbation to a broader range of loss functions.** The privacy guarantees of objective perturbation require the loss function to have bounded gradient. Our proposed framework extends the Approximate Minima Perturbation framework of Iyengar et al. (2019) to take any smooth loss function as a blackbox, then algorithmically ensure that it has bounded gradient. We also provide a **computational guarantee** $O(n \log n)$ on the running time of this algorithm, in contrast to the $O(n^2)$ complexity of DP-SGD for achieving information-theoretic limits.

## 1.2 A Short History of DP Learning

Differentially private learning dates back to Chaudhuri et al. (2011), which extended the output perturbation method of Dwork et al. (2006) to classification algorithms and also introduced objective perturbation. In its first public appearance, objective perturbation required gamma-distributed noise; Kifer et al. (2012) provided a refined analysis of the mechanism with Gaussian noise, which is the entry point into our work.

Differentially private stochastic gradient descent (DP-SGD) (Song et al., 2013; Bassily et al., 2014; Abadi et al., 2016) brought DP into the fold of modern machine learning, allowing private training of models with arbitrarily complex loss landscapes that can scale to enormous datasets. DP-SGD adds Gaussian noise at every iteration to an aggregation of clipped gradients, and thus privacy analysis for DP-SGD often boils down to finding tight composition bounds (of the subsampled Gaussian mechanism).

The initial version of DP-SGD based on the standard strong composition (Bassily et al., 2014) is not quite practical, but that has changed, thanks to a community-wide effort in developing modern numerical privacy accounting tools in the past few years. These include the moments accountant that composes Renyi DP functions (Abadi et al., 2016; Wang et al., 2019; Mironov et al., 2019) and the Fourier accountant (also known as PLV or PLD accountants) that directly compose the *privacy profiles* (Sommer et al., 2019; Koskela et al., 2020; Gopi et al., 2021; Zhu et al., 2022). It is safe to conclude that the numerically computed privacy loss of DP-SGD using these modern tools is now very precise.

---

[1]"Honest" hyperparameter tuning is a term coined by Mohapatra et al. (2022).

[2]The concurrent work of Agarwal et al. (2023) has independently identified this bug as well.

Because DP-SGD releases each intermediate model, the algorithm can stop after any number of iterations and simply accumulates privacy loss as it goes. In contrast, the privacy guarantees of objective perturbation hold only when the output of the mechanism is the *exact* minima of the perturbed objective. This requirement is at odds with practical convex optimization frameworks which typically use first-order methods to search for an approximate solution.

To remedy this, Iyengar et al. (2019) proposed an approach to *approximately* minimize a perturbed objective function while maintaining privacy. Approximate Minima Perturbation (AMP) was introduced as a tractable alternative to objective perturbation whose privacy guarantees permit the output to be an approximate (rather than exact) solution to the perturbed minimization problem. In this paper we extend AMP to a broader range of loss functions; Algorithm 1 can be viewed as a special case of AMP with a transformation of the loss function.

## 2 Preliminaries

### 2.1 Differential Privacy

Differential privacy (DP) (Dwork et al., 2006) offers provable privacy protection by restricting how much the output of a randomized algorithm can leak information about a single data point.

DP requires a notion of how to measure similarity between datasets. We say that datasets $Z$ and $Z'$ are neighboring datasets (denoted $Z \simeq Z'$) if they differ by exactly one datapoint $z$, i.e. $Z' = Z \cup \{z\}$ or $Z' = Z \setminus \{z\}$ for some data entry $z$.

**Definition 2.1** (Differential privacy). A mechanism $\mathcal{M} : \mathcal{Z} \to \mathcal{R}$ satisfies $(\epsilon, \delta)$-differential privacy if for all neighboring datasets $Z, Z' \in \mathcal{Z}$ and output sets $S \subseteq \mathcal{R}$,

$$\Pr\left[\mathcal{M}(Z) \in S\right] \le e^\epsilon \Pr\left[\mathcal{M}(Z') \in S\right] + \delta.$$

When $\delta > 0$, $\mathcal{M}$ satisfies *approximate DP*. When $\delta = 0$, $\mathcal{M}$ satisfies the stronger notion of *pure DP*.

We say that $\mathcal{M}$ is tightly $(\epsilon, \delta)$-DP if there is no $\delta' < \delta$ for which $\mathcal{M}$ would be $(\epsilon, \delta')$-DP.

In what follows, we overview two different styles of achieving DP guarantees: one via hockey-stick divergence, and the other via Rényi divergence.

#### 2.1.1 DP via hockey-stick divergence

**Definition 2.2** (Hockey-stick divergence). Denote $[x]_+ = \max\{0, x\}$ for $x \in \mathbb{R}$. For $\alpha > 0$ the hockey-stick divergence $H_\alpha$ from a distribution $P$ to a distribution $Q$ is defined as

$$H_\alpha(P||Q) = \int \left[P(x) - \alpha \cdot Q(x)\right]_+ \, \mathrm{d}x.$$

Now (with some abuse of notation) we will discuss how to bound the hockey-stick divergence between distributions $\mathcal{M}(Z)$ and $\mathcal{M}(Z')$ via the concept of *privacy profiles*.

**Definition 2.3** (Privacy profiles Balle et al., 2018). The privacy profile $\delta_\mathcal{M}(\epsilon)$ of a mechanism $\mathcal{M}$ is defined as

$$\delta_\mathcal{M}(\epsilon) := \max_{Z \simeq Z'} H_{\mathrm{e}^\epsilon}\left(\mathcal{M}(Z)||\mathcal{M}(Z')\right).$$

Tight $(\epsilon, \delta)$-DP bounds can then be obtained as follows.

**Lemma 2.4** (Zhu et al., 2022, Lemma 5). *Mechanism $\mathcal{M}$ satisfies $(\epsilon, \delta)$-DP if any only if $\delta \ge \delta_\mathcal{M}(\epsilon)$.*

*Dominating pairs* of distributions are useful for bounding the hockey-stick divergence $H_{\mathrm{e}^\epsilon}\left(\mathcal{M}(Z)||\mathcal{M}(Z')\right)$ accurately and, in particular, for obtaining tight bounds for compositions.

**Definition 2.5** (Zhu et al. 2022). A pair of distributions $(P, Q)$ is a *dominating pair* of distributions for mechanism $\mathcal{M} : \mathcal{Z} \to \mathcal{R}$ if for all neighboring datasets $Z$ and $Z'$ and for all $\alpha > 0$,

$$H_\alpha(\mathcal{M}(Z)||\mathcal{M}(Z')) \le H_\alpha(P||Q).$$

### 2.1.2 DP via Rényi divergence

**Definition 2.6.** (Rényi divergence.) Let $\alpha > 0$. For $\alpha \neq 1$, the Rényi divergence $D_\alpha$ from distribution $P$ to distribution $Q$ is defined as

$$D_\alpha(P||Q) = \frac{1}{\alpha - 1} \log \mathbb{E}_{x \sim Q} \left[ \left( \frac{P(x)}{Q(x)} \right)^\alpha \right].$$

When $\alpha = 1$, Rényi divergence reduces to the Kullback–Leibler (KL) divergence:

$$D_1(P||Q) = \mathbb{E}_{x \sim P} \left[ \log \left( \frac{P(x)}{Q(x)} \right) \right].$$

Rényi differential privacy (RDP) is a relaxation of pure DP ($\delta = 0$) based on Rényi divergence.

**Definition 2.7** (Rényi differential privacy)**.** A mechanism $\mathcal{M} : \mathcal{Z} \rightarrow \mathcal{R}$ satisfies $(\alpha, \epsilon)$-Rényi differential privacy if for all neighboring datasets $Z, Z' \in \mathcal{Z}$,

$$D_\alpha\big(\mathcal{M}(Z) \,||\, \mathcal{M}(Z')\big) \leq \epsilon,$$

RDP implies $(\epsilon, \delta)$-DP for any $0 < \delta \leq 1$ with $\epsilon = \min_{\alpha > 1}\{\epsilon(\alpha) + \log(1/\delta)/(\alpha - 1)\}$. Tighter but more complex conversion formulae were derived by Balle et al. (2020) and Canonne et al. (2020), which we adopt numerically in our experiments whenever approximate DP is needed.

## 2.2 Differentially Private Empirical Risk Minimization

We have a dataset $Z \in \mathcal{Z}$ and a loss function $\ell(\theta; z)$; we want to solve problems of the form

$$\hat{\theta} = \arg\min_{\theta \in \Theta} \sum_{z \in Z} \ell(\theta; z) + r(\theta),$$

where $z = (x, y) \in \mathcal{X} \times \mathcal{Y}$ is a data point and $r(\theta)$ is a regularization term. The feature space is $\mathcal{X} \subseteq \mathbb{R}^d$ and the label space is $\mathcal{Y} \subseteq \mathbb{R}$. We will assume that $||x||_2 \leq 1$ and $|y| \leq 1$.

This work focuses on unconstrained convex generalized linear models (GLMs): we require that $\ell(\theta)$ and $r(\theta)$ are convex and twice-differentiable and that $\Theta = \mathbb{R}^d$. The loss function is assumed to have GLM structure of the form $\ell(\theta; z) = f(x^T\theta; y)$.

**Objective Perturbation** Construct the perturbed objective function by sampling $b \sim \mathcal{N}(0, \sigma^2 I_d)$:

$$\mathcal{L}^P(\theta; Z, b) = \sum_{z \in Z} \ell(\theta; z) + \frac{\lambda}{2} ||\theta||_2^2 + b^T\theta.$$

The objective perturbation mechanism (ObjPert) outputs $\hat{\theta}^P(Z) = \arg\min_{\theta \in \Theta} \mathcal{L}^P(\theta; Z, b)$.

**Theorem 2.8** (DP guarantees of objective perturbation (Kifer et al., 2012))**.** *Let $\ell(\theta; z)$ be convex and twice-differentiable such that $||\nabla\ell(\theta; z)||_2 \leq L$ and $\nabla^2\ell(\theta; z) \prec \beta I_d$ for all $\theta \in \Theta$ and $z \in \mathcal{X} \times \mathcal{Y}$. Then objective perturbation satisfies $(\epsilon, \delta)$-DP when $\lambda \geq \frac{2\beta}{\epsilon}$ and $\sigma \geq \frac{L\sqrt{8\log(2/\delta)+4\epsilon}}{\epsilon}$.*

**Differentially Private Gradient Descent** DP-SGD is a differentially private version of stochastic gradient descent which ensures privacy by clipping the per-example gradients at each iteration before aggregating them and adding noise to the result. The update rule at iteration $t$ is given by

$$\theta_{t+1} = \theta_t - \eta_t \left( \sum_{z \in B_t} \texttt{clip}\big(\nabla\ell(\theta_t; z)\big) + \mathcal{N}(0, \sigma^2 I_d) \right),$$

where $\eta_t$ is the learning rate at iteration $t$, $B_t$ is the current batch at iteration $t$, $\sigma$ is the noise scale, and $\texttt{clip}$ is a function that bounds the norm of the per-example gradients.

# 3 Analytical Tools

Existing privacy guarantees of the objective perturbation mechanism (Chaudhuri et al., 2011; Kifer et al., 2012) pre-date modern privacy accounting tools such as Rényi differential privacy and privacy profiles. In this section, we present two new privacy analyses of objective perturbation: an $(\epsilon, \delta)$-DP bound based on privacy profiles, and an RDP bound.

## 3.1 Approximate DP Bound

**Theorem 3.1** (Approximate DP guarantees of objective perturbation for GLMs)**.** *Consider a loss function $\ell(\theta; z) = f(x^T \theta; y)$ with GLM structure. Suppose that $f$ is $\beta$-smooth and $||\nabla \ell(\theta; z)||_2 \leq L$ for all $\theta \in \Theta$ and $z \in \mathcal{X} \times \mathcal{Y}$. Fix $\lambda > \beta$. Let $\epsilon \geq 0$ and let $\tilde{\epsilon} = \epsilon - \log\left(1 - \frac{\beta}{\lambda}\right)$, $\hat{\epsilon} = \epsilon - \log\left(1 - \frac{\beta}{\lambda}\right) - \frac{L^2}{2\sigma^2}$, and let $P$ and $Q$ be the density functions of $\mathcal{N}(L, \sigma^2)$ and $\mathcal{N}(0, \sigma^2)$, respectively. Objective perturbation satisfies $\left(\epsilon, \delta(\epsilon)\right)$-DP for*

$$\delta(\epsilon) = \begin{cases} 2 \cdot H_{e^{\tilde{\epsilon}}}\left(P||Q\right), & \text{if } \hat{\epsilon} \geq 0, \\ (1 - e^{\hat{\epsilon}}) + e^{\hat{\epsilon}} \cdot 2 \cdot H_{e^{\frac{L^2}{\sigma^2}}}\left(P||Q\right), & \text{otherwise.} \end{cases} \tag{3.1}$$

Notice that we can express (3.1) analytically using (B.1). To obtain the bound (3.1) we repeatedly use the fact that the privacy loss random variable (PLRV) determined by the distributions $\mathcal{N}(1, \sigma^2)$ and $\mathcal{N}(0, \sigma^2)$ is distributed as $\mathcal{N}(\frac{1}{2\sigma^2}, \frac{1}{\sigma^2})$. As the upper bound (3.1) is obtained using a PLRV that is a certain scaled and shifted half-normal distribution, we can also find certain scaled and shifted half-normal distributions $P$ and $Q$ which give the dominating pair of distributions for the objective perturbation mechanism such that the hockey-stick divergence between $P$ and $Q$ is exactly the upper bound (3.1) for all $\epsilon$ (shown in the appendix).

## 3.2 Rényi Differential Privacy Bound

If our sole objective is to obtain the tightest possible approximate DP bounds for objective perturbation, we can stop at Theorem 3.1! Directly calculating the privacy profiles of objective perturbation using the hockey-stick divergence, as in the previous section, will achieve this goal (until more privacy accounting tools come along).

In this section we turn instead to Rényi differential privacy, a popular relaxation of pure differential privacy ($\delta = 0$) which avoids the "catastrophic privacy breach" possibility permitted by approximate DP ($\delta > 0$). Below, we present an RDP guarantee for objective perturbation.

**Theorem 3.2** (RDP guarantees of objective perturbation for GLMs)**.** *Consider a loss function $\ell(\theta; z) = f(x^T \theta; y)$ with GLM structure. Suppose that $f$ is $\beta$-smooth and $||\nabla \ell(\theta; z)||_2 \leq L$ for all $\theta \in \Theta$ and $z \in \mathcal{X} \times \mathcal{Y}$. Fix $\lambda > \beta$. Objective perturbation satisfies $(\alpha, \epsilon)$-RDP for any $\alpha > 1$ with*

$$\epsilon = -\log\left(1 - \frac{\beta}{\lambda}\right) + \frac{L^2}{2\sigma^2} + \frac{1}{\alpha - 1} \log \mathbb{E}_{X \sim \mathcal{N}\left(0, \frac{L^2}{\sigma^2}\right)}\left[e^{(\alpha - 1)|X|}\right].$$

*For $\alpha = 1$, the RDP bound holds with*

$$\epsilon = -\log\left(1 - \frac{\beta}{\lambda}\right) + \frac{L^2}{2\sigma^2} + \log \mathbb{E}_{X \sim \mathcal{N}\left(0, \frac{L^2}{\sigma^2}\right)}\left[e^{|X|}\right].$$

One of our main motivations for improving the privacy analysis of objective perturbation comes from the observation that it can be competitive to DP-SGD when the privacy cost of hyperparameter tuning is included in the privacy budget. As the tightest results for DP hyperparameter tuning are stated in terms of RDP (Papernot & Steinke, 2022), in our experiments we use RDP bounds of objective perturbation to get a clear understanding of the differences in the privacy-utility trade-offs between these two approaches.

*Remark* 3.3. Privacy profile and RDP bounds (such as Theorems 3.1 and 3.2) are unified in the sense that they are both based on a certain bound of the PLRV $\epsilon_{Z,Z'}$ (Definition K.4) for a fixed pair of datasets $Z, Z'$. From Definitions 2.2 and 2.7 we see that for $\epsilon \in \mathbb{R}$, the hockey-stick divergence is

$$H_{e^\epsilon}\left(\mathcal{M}(Z) \,||\, \mathcal{M}(Z')\right) = \mathbb{E}_{\theta \sim \mathcal{M}(Z)}\left[1 - e^{\epsilon - \epsilon_{Z,Z'}(\theta)}\right]_+,$$

and for $\alpha > 1$ we have that the Rényi divergence is

$$D_\alpha\left(\mathcal{M}(Z) \,||\, \mathcal{M}(Z')\right) = \frac{1}{\alpha - 1} \log \mathbb{E}_{\theta \sim \mathcal{M}(Z)}\left[e^{\alpha \epsilon_{Z,Z'}(\theta)}\right].$$

### 3.3 Distance to Optimality

How close to optimal are the bounds of Theorems 3.1 and 3.2 ? We can in fact show that the Gaussian mechanism is a special case of the objective perturbation mechanism — thereby providing a lower bound on its approximate DP and RDP.

*Example* 3.4. *Consider the loss function $\ell(\theta; x) = x^T \theta$ and choose neighboring datasets $X = \{x\}$ and $X' = \emptyset$, for some $x \in \mathbb{R}^d$. Fix $\lambda > 0$ and sample $b \sim \mathcal{N}(0, \sigma^2 I_d)$. Then the objective perturbation mechanism solves*

$$\hat{\theta}^P(X) = \underset{\theta \in \mathbb{R}^d}{\arg\min} \; x^T \theta + \frac{\lambda}{2}||\theta||_2^2 + b^T \theta = -\frac{1}{\lambda}(x + b),$$

$$\hat{\theta}^P(X') = \underset{\theta \in \mathbb{R}^d}{\arg\min} \; \frac{\lambda}{2}||\theta||_2^2 + b^T \theta = -\frac{1}{\lambda}b.$$

*Observe that $\hat{\theta}^P(X) \sim \mathcal{N}(-\frac{1}{\lambda}x, \frac{\sigma^2}{\lambda^2}I_d)$ and $\hat{\theta}^P(X') \sim \mathcal{N}(0, \frac{\sigma^2}{\lambda^2}I_d)$. Following the problem setting described in Theorem 2.8, we have that $||x||_2 = ||\nabla \ell(\theta; x)||_2 \leq L$. In this case, objective perturbation reduces directly to the Gaussian mechanism with sensitivity $\Delta_f = \frac{L}{\lambda}$ and noise scale $\frac{\sigma}{\lambda}$.*

*Corollary* 3.5. *As a consequence of Example 3.4 and the scaling invariance of the hockey-stick divergence, for all $\alpha > 1$ we have the following:*

$$H_\alpha\big(\hat{\theta}^P(Z) \,\|\, \hat{\theta}^P(Z')\big) \geq H_\alpha\big(\mathcal{N}(0, \tfrac{\sigma^2}{\lambda^2}) \,\|\, \mathcal{N}(\tfrac{L}{\lambda}, \tfrac{\sigma^2}{\lambda^2})\big) = H_\alpha\big(\mathcal{N}(0, \sigma^2) \,\|\, \mathcal{N}(L, \sigma^2)\big).$$

The argument works the same for the Rényi divergence $D_\alpha$ which is similarly scaling-invariant. Corollary 3.5 implies that we can measure the tightness of the bounds given in Theorems 3.1 and 3.2 by comparing them to the tight bounds of the Gaussian mechanism (B.1) with sensitivity $\Delta_f = L$ and noise scale $\sigma$.

This means that in Figure 1, the hockey-stick divergence of the Gaussian mechanism is a lower bound on the hockey-stick divergence for objective perturbation. While our hockey-stick divergence bound is unsurprisingly a bit tighter than the RDP bound for objective perturbation, we see that both significantly improve over the classic $(\epsilon, \delta)$-DP bounds of Kifer et al. (2012).

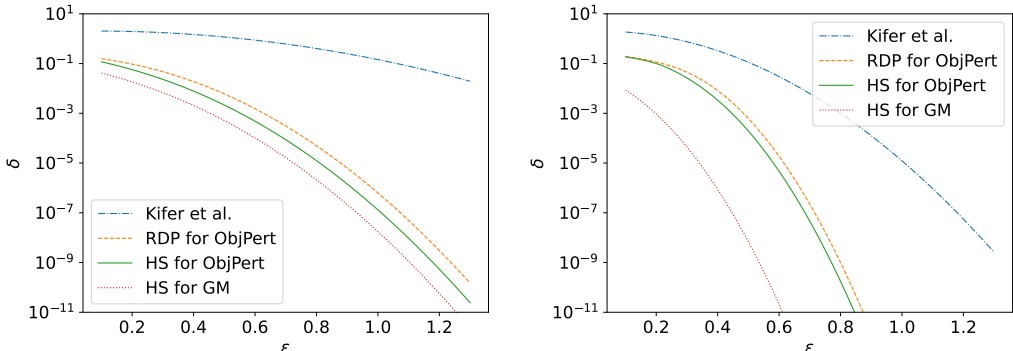

Figure 1: Comparison of different $(\epsilon, \delta)$-bounds for objective perturbation: the $(\epsilon, \delta)$-bound by Kifer et al. (2012) given in Thm. 2.8, the RDP bound of Thm. 3.2, the approximate DP bound of Thm. 3.1 using the hockey-stick divergence and the approximate DP lower bound obtained using the hockey-stick divergence and Cor. 3.5. Left: $\sigma = 5.0$, $\beta = 1.0$ and $\lambda = 20.0$. Right: $\sigma = 10.0$, $\beta = 1.0$ and $\lambda = 5.0$.

*Remark* 3.6. The RDP and approximate DP bounds in this section require a careful analysis of the dependence between the noise vector $b$ and the private minimizer $\theta^P$. In the appendix, we show how the GLM assumption simplifies this issue.

## 4 Computational Tools

In this section we present Algorithm 1, which extends the Approximate Minima Perturbation of Iyengar et al. (2019) to handle loss functions with unbounded gradient.

**Approximate minima** The privacy guarantees of objective perturbation hold only when its output is an *exact* minimizer of the perturbed objective. Approximate Minima Perturbation (AMP) (Iyengar et al., 2019) addresses this issue by finding an approximate minimizer to the perturbed objective, then privately releases this approximate minimizer with the Gaussian mechanism.

**Gradient clipping** DP-SGD requires no *a priori* bound on the gradient of the loss function; at each iteration, the algorithm clips the per-example gradients above a pre-specified threshold in order to bound the gradient norms. We extend this same technique to objective perturbation.

Given a loss function $\ell(\theta; z)$ and a clipping threshold $C$, we can transform the gradient of $\ell(\theta; z)$ as follows:

$$\nabla \ell_C(\theta; z) = \begin{cases} \nabla \ell(\theta; z) & \text{if } ||\nabla \ell(\theta; z)||_2 \leq C, \\ \frac{C}{||\nabla \ell(\theta; z)||_2} \nabla \ell(\theta; z) & \text{if } ||\nabla \ell(\theta; z)||_2 > C. \end{cases}$$

Then we can define the aggregation of clipped gradients as $\nabla \mathcal{L}_C(\theta; Z) = \sum_{z \in Z} \nabla \ell_C(\theta; z)$.

The aggregation of clipped gradients $\nabla \mathcal{L}_C(\theta; Z)$ corresponds to an implicit "clipped-gradient" objective function $\mathcal{L}_C(\theta; Z)$. For convex GLMs, Song et al. (2020) define this function precisely and show that it retains the convexity and GLM structure of the original objective function $\mathcal{L}(\theta; Z)$. We furthermore demonstrate that this function retains the same bound $\beta$ on the Lipschitz smoothness (Theorem E.3).

Algorithm 1 extends the privacy guarantees of AMP (Iyengar et al., 2019) to loss functions with unbounded gradient. Notice that for smooth loss functions with gradient norm bounded by $L$, we can set $C = L$ in order to recover Approximate Minima Perturbation.

---

**Algorithm 1** Computational Objective Perturbation

---

**Input:** dataset $Z$; noise levels $\sigma, \sigma_{\text{out}}$; $\beta$-smooth loss function $\ell(\cdot)$ ; regularization strength $\lambda$; gradient norm threshold $\tau$; clipping threshold $C$.
1. Construct the set of clipped-gradient loss functions $\{\ell_C(\theta; z) : z \in Z\}$.
2. Sample $b \sim \mathcal{N}(0, \sigma^2 I_d)$.
3. Let $\mathcal{L}_C^P(\theta; Z, b) = \sum_{z \in Z} \ell_C(\theta; z) + \frac{\lambda}{2}||\theta||_2^2 + b^T \theta$.
4. Solve for $\tilde{\theta}$ such that $||\nabla \mathcal{L}_C^P(\tilde{\theta}; Z)||_2 \leq \tau$.
**Output:** $\tilde{\theta}^P = \tilde{\theta} + \mathcal{N}(0, \sigma_{\text{out}}^2 I_d)$.

---

*Theorem* 4.1 (RDP guarantees of Algorithm 1). *Consider a loss function $\ell(\theta; z) = f(x^T \theta; y)$ with GLM structure, such that $f$ is $\beta$-smooth. Fix $\lambda > \beta$. Algorithm 1 satisfies $(\alpha, \epsilon)$-RDP for any $\alpha > 1$ with*

$$\epsilon \leq -\log\left(1 - \frac{\beta}{\lambda}\right) + \frac{C^2}{2\sigma^2} + \frac{1}{\alpha - 1} \log \mathbb{E}_{X \sim \mathcal{N}\left(0, \frac{C^2}{\sigma^2}\right)}\left[e^{(\alpha-1)|X|}\right] + \frac{2\tau^2 \alpha}{\sigma_{out}^2 \lambda^2}.$$

*Remark* 4.2. Gradient clipping aside, our proof of Theorem 4.1 takes a different tack than the proof of Theorem 1 (for AMP) in Iyengar et al. (2019). We observe that Algorithm 1 is essentially an adaptive composition of the objective perturbation mechanism and the Gaussian mechanism. We can write $\tilde{\theta} = \theta^P + (\tilde{\theta} - \theta^P)$ to see that we are releasing two quantities: $\theta^P$ (with objective perturbation) and the difference $\tilde{\theta} - \theta^P$ (with the Gaussian mechanism). Algorithm 1 stops iterating only after the gradient norm $||\nabla \mathcal{L}_C^P(\tilde{\theta}; Z)||_2$ is below the threshold $\tau$. This along with the $\lambda$-strong convexity of the objective function $\nabla \mathcal{L}_C^P(\theta; Z)$ ensures a bound on the $\ell_2$-sensitivity of the difference $\tilde{\theta} - \theta^P$, so that we can apply the Gaussian mechanism.

## 4.1 Computational Guarantee

To achieve the optimal excess risk bounds for DP-ERM in the convex setting, DP-SGD clocks in at a hefty $O(n^2)$ gradient evaluations (Bassily et al., 2014). It has been an open problem to obtain optimal DP-ERM algorithms that runs in subquadratic time (Kulkarni et al., 2021). One of our contributions is to show that when we further restrict to smooth GLM-losses (so ObjPert is applicable) Algorithm 1 can achieve the same optimal rate with only $O(n \log n)$ gradient evaluations.

A formal claim and proof that Algorithm 1 — with appropriately chosen parameters — achieves the optimal rate is deferred to Appendix H. The analysis is largely the same as those in (Kifer et al., 2012) but with the bug fixed (details in Appendix G) by adding a GLM assumption.

The improved computational complexity is due to that we can apply any off-the-shelf optimization algorithms to solve Step 4 of Algorithm 1. Observe that $\mathcal{L}^P(\theta; Z, b)$ has a finite-sum structure, we can employ the Stochastic Averaged Gradient (SAG) method (Schmidt et al., 2017) which halts in $O(n \log n)$ with high probability. Details are provided in Appendix F.

## 5 Empirical Evaluation

In this section we evaluate Algorithm 1 against two baselines: "dishonest" DP-SGD and "honest" DP-SGD. Dishonest DP-SGD does not account for the privacy cost of hyperparameter tuning; honest DP-SGD follows the private selection algorithm and RDP bounds from Papernot & Steinke (2022).

Our experimental design includes some guidelines in order to make it a fair fight. One of the strengths of Algorithm 1 that we advocate for is its blackbox optimization. Whereas DP-SGD consumes privacy budget for testing each set of hyperparameter candidates, an advantage of approximate minima perturbation is that the privacy guarantees are independent of the choice of optimizer used to solve for $\tilde{\theta}$. We can therefore test out any number of optimization hyperparameters for Algorithm 1 *at no additional privacy cost*, provided that these parameters are independent of the privacy guarantee (e.g. learning rate, batch size). More specifically, once the loss function is perturbed with the noise $b$ in Algorithm 1, any $\tilde{\theta}$ that satisfies the convergence guarantees with the tolerance parameter $\tau$ will have the RDP-guarantees of Theorem 4.1 and therefore we are free to also carry out tuning of the optimization algorithm without an additional privacy cost.

Because we are interested in measuring the effect of the privacy cost of hyperparameter tuning, we tune only the learning rate which does not affect the privacy guarantee of the base algorithm. This isolates the effect of hyperparameter tuning as we will need to appeal to Papernot & Steinke (2022) to get valid DP bounds for DP-SGD, but Algorithm 1 enjoys hyperparameter tuning "for free".

The following table summarizes the optimization-related parameters for all three methods.

| | Dishonest DP-SGD | Honest DP-SGD | Algorithm 1 |
|---|---|---|---|
| clipping | $C = \sqrt{2}$ | $C = \sqrt{2}$ | $C = \sqrt{2}$ |
| learning rate | $\log(10^{-8}, 10^{-1})$ | $\log(10^{-8}, 10^{-1})$ | linear$(.08, .5)$ |
| grid size | $s = 10$ | $K \sim \text{Poisson}(\mu)$ s.t. $\Pr[K > s] = 0.9$ | $s = 10$ |
| optimizer | Adam | Adam | L-BFGS |
| convergence | after $T$ iterations | after $T$ iterations | $\|\nabla\mathcal{L}(\tilde{\theta})\|_2 \leq \tau$ |

The choice of $C = \sqrt{2}$ is a natural value for logistic regression in that $\|\nabla\ell(\theta, z)\| \leq \sqrt{2}$ for all $\theta, z$ due to data-preprocessing and the bias term. Dishonest DP-SGD selects $s = 10$ learning rate candidates evenly log-spaced from the range of values between $10^{-8}$ and $10^{-1}$. Honest DP-SGD selects learning rate candidates from the same range of values, but with granularity determined by a random variable $K$ sampled from the Poisson distribution $\text{Poisson}(\mu)$. We select $\mu$ so that with 90% probability, $K$ is larger than the grid size $s$ used for dishonest DP-SGD, resulting in $\mu \approx 15.4$.

We use the Adam optimizer for both honest and dishonest DP-SGD. For Algorithm 1 we use the L-BFGS optimizer whose second-order behavior allows us to get within a smaller distance to optimal (as required by Algorithm 1).

For DP-SGD we set the subsampling ratio such that the expected batch size is 256 and we run for 60 "epochs". We calculate the number of iterations as $T = 60 \cdot \texttt{num\_batches}$, where $\texttt{num\_batches}$ is the number of batches in the training dataset (we pass the train loader through the `Opacus` privacy engine, so the size of each batch is random). To calibrate the scale of the noise for DP-SGD, we use the analytical moments accountant (Wang et al., 2019) with Poisson sampling (Zhu & Wang, 2019; Mironov et al., 2019).

The parameters specific to AMP are $\sigma_{out}$, the noise scale for the output perturbation step; and $\tau$, the gradient norm bound. A larger $\tau$ will improve our computational cost, though our approximate minimizer $\tilde{\theta}$ will be farther away from the true minimizer $\theta^P$. We can achieve a smaller $\tau$ by choosing a larger $\sigma_{out}$, but this will mean that our release of $\tilde{\theta}$ will be noisier. In our experiments we fix $\tau = 0.01$ and $\sigma_{out} = 0.15$.

The privacy parameters of objective perturbation are the noise scale $\sigma$ and the regularization strength $\lambda$. Balancing these parameters is a classic exercise in bias-variance trade-off. A larger $\sigma$ will allow us to use less regularization, but if $\sigma$ is too large then we risk adding too much noise to the objective function and hurting utility.

Our strategy is to find the smallest possible $\lambda$ such that $\sigma$ isn't too large. To quantify when $\sigma$ is "too large", we use the Gaussian mechanism as a reference point: the noise scale $\sigma$ for objective perturbation shouldn't be too much larger than the noise scale $\sigma_G$ for the Gaussian mechanism. Let's say that the Gaussian mechanism with noise scale $\sigma_G$ satisfies $(\epsilon, \delta)$-DP, then we want our $\sigma$ for $(\epsilon, \delta)$-DP objective perturbation to satisfy $\sigma \leq f\sigma_G$ for some small constant factor $f$. In our experiments, we set $f = 1.3$.

For objective perturbation, we can thus select the privacy parameters $\sigma$ and $\lambda$ using fixed values (e.g., $\epsilon, \delta, \sigma_G$) that are independent of the data. Likewise, the choices of $\sigma_{out}$ and $\tau$ are fixed across all datasets. This is noteworthy since $\sigma, \lambda, \sigma_{out}$ and $\tau$ each have an effect on the privacy guarantee, outside of the blackbox algorithm. Tuning these parameters on the data would require us to use the private selection algorithm.

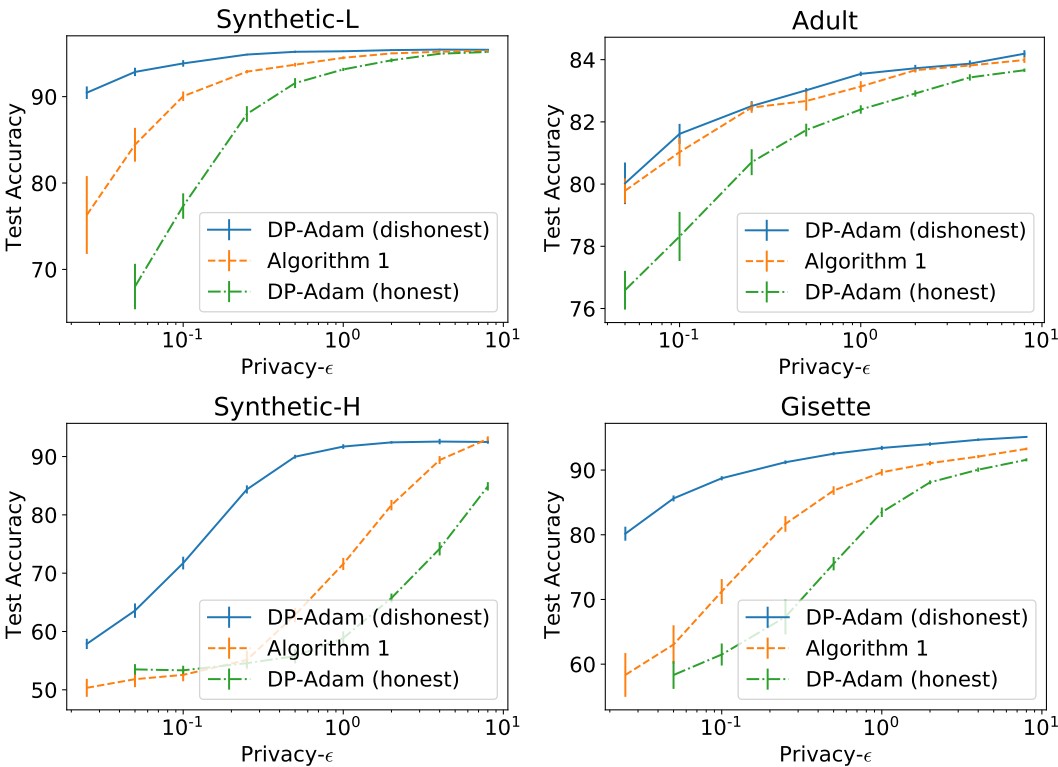

Figure 2: Comparison of Algorithm 1 against honest and dishonest DP-SGD baselines, varying $\epsilon \in \{0.025, 0.05, 0.1, 0.25, 0.5, 1.0, 2.0, 4.0, 8.0\}$ and fixing $\delta = 10^{-5}$. On all three methods, we train the model for each learning rate on its grid (see Table 5) and report the test accuracy for the best learning rate on the grid. Results are averaged over 10 trials and the error bars on both sides of the mean values depict 1.96 times the standard error, giving the asymptotic 95% coverage.

We evaluate our methods for binary classification on the Adult, Synthetic-L, Synthetic-H and Gisette datasets provided by Iyengar et al. (2019). We normalize each row $x_i$ to have unit $\ell_2$-norm. Note

that the assignment $x_i \leftarrow \frac{x_i}{||x_i||_2}$ doesn't require expending any privacy budget as each data point is transformed only by its own per-sample norm.

Table 1: Synthetic-L

|  | Dishonest | Alg 1 | Honest |
|---|---|---|---|
| $\epsilon = 0.1$ | 93.85% | 90.05% | 77.34% |
| $\epsilon = 1$ | 95.25 | 94.50% | 93.15% |
| $\epsilon = 8$ | 95.43% | 95.30% | 95.17 % |

Table 2: Adult

|  | Dishonest | Alg 1 | Honest |
|---|---|---|---|
| $\epsilon = 0.1$ | 81.61% | 81.37% | 78.32% |
| $\epsilon = 1$ | 83.54% | 83.18% | 82.40% |
| $\epsilon = 8$ | 84.19% | 83.99% | 83.66% |

Results from Figure 5 in numerical format for the low-dimensional datasets, Synthetic-L and Adult. For $\epsilon = 0.1$, these can be cross-referenced with the results in Fig. 3 from (Iyengar et al., 2019).

The experimental results shown in Figure 5, Table 1 and Table 2 paint a consistent picture. While dishonest DP-SGD is clearly the best-performing algorithm, when we account for the cost of hyperparameter tuning then Algorithm 1 can typically best honest DP-SGD. This effect is especially pronounced under small $\epsilon$, for which diverting some of the limited privacy budget to hyperparameter tuning could be more impactful.

**Is it fair?** Our experimental design aims to fairly compare ObjPert to DP-SGD. One limitation, however, is that the state-of-the-art tools for private hyperparameter tuning from Papernot & Steinke (2022) are RDP bounds — and RDP is *not* state-of-the-art for DP-SGD privacy accounting. At this moment, the tighest privacy accounting tool for DP-SGD is the PRV accountant from Gopi et al. (2021), which is the counterpart to our privacy profiles analysis for ObjPert (Theorem 3.1). Unfortunately, even though dishonest DP-SGD would benefit from using the PRV accountant, for private hyperparameter tuning we would then have to use the sub-optimal private selection bounds for approximate DP from Liu & Talwar (2019). In our experiments we therefore use RDP-based privacy accounting for both ObjPert and DP-SGD. Comparing DP-SGD with numerical composition against ObjPert with Theorem 3.1 will have to wait until more private selection tools are available.

One might also object that by tuning only the learning rate for DP-SGD, we didn't explore the full range of hyperparameters relevant to DP-SGD's performance. While tuning additional hyperparameters such as the batch size and number of epochs could benefit dishonest DP-SGD, it would likely worsen the privacy-utility tradeoff for honestly-tuned DP-SGD due to the increased privacy cost of the hyperparameter tuning algorithm from Papernot & Steinke (2022).

## 6    Conclusion

One point that we really wanted to drive home is that while DP-SGD works extraordinarily well across a wide variety of problem settings, it's not necessarily the best solution for *every* problem setting. But at the same time, DP-SGD has received the benefit of an enormous amount of attention that other DP learning algorithms haven't received. A goal of our paper was to hone in on a particular problem setting and give a different algorithm the same star treatment.

Objective perturbation now boasts two new privacy analyses. One is an improved $(\epsilon, \delta)$-DP analysis based on bounding the hockey-stick divergence. The other is an RDP analysis which allows us to fairly compare objective perturbation against DP-SGD — the workhorse of differentially private learning — with honest hyperparameter tuning. We've also expanded the approximate minima perturbation algorithm of Iyengar et al. (2019) in order to encompass a broader range of loss functions which need not have bounded gradient. Our algorithm moreover can be used in conjunction with SVRG to guarantee a running time of $O(n \log n)$ to achieve the optimal excess risk bounds, improving on the $O(n^2)$ computational guarantee of DP-SGD.

### Acknowledgments

The work was partially supported by NSF Award #2048091.

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

# Appendix

## Table of Contents

# A   Notation

Denote the following:

- $\mathcal{L}(\theta) = \sum\limits_{i=1}^{n} \ell_i(\theta),$

- $\mathcal{L}_\lambda(\theta) = \mathcal{L}(\theta) + \frac{\lambda}{2}||\theta||_2^2,$

- $\mathcal{L}^P(\theta) = \mathcal{L}(\theta) + \frac{\lambda}{2}||\theta||_2^2 + b^T\theta, \ \ b \sim \mathcal{N}(0, \sigma^2 I_d),$

- $\theta^* = \underset{\theta\in\mathbb{R}^d}{\arg\min}\,\mathcal{L}(\theta),$

- $\theta_\lambda^* = \underset{\theta\in\mathbb{R}^d}{\arg\min}\,\mathcal{L}_\lambda(\theta),$

- $\theta^P = \underset{\theta\in\mathbb{R}^d}{\arg\min}\,\mathcal{L}^P(\theta),$

- $\tilde{\theta}$ satisfies $||\nabla\mathcal{L}^P(\tilde{\theta})||_2 \le \tau,$

- $\tilde{\theta}^P = \tilde{\theta} + b_2 \ , b_2 \sim \mathcal{N}(0, \sigma_{out}^2 I_d).$

We take $||\mathcal{X}||$ to be the size of the data domain $\mathcal{X}$, i.e. $||\mathcal{X}|| = \sup_{x\in\mathcal{X}} ||x||$. For conciseness of presentation we sometimes drop the dataset $Z$ from the notation, e.g. we abbreviate $\mathcal{L}(\theta; Z)$ as $\mathcal{L}(\theta)$.

Sometimes (especially in the proofs that follow) we will abuse notation by overloading a function with its output, e.g. $\theta^P$ is the output of objective perturbation and $\theta^P(Z)$ is the objective perturbation mechanism.

# B   Warm-up: Gaussian Mechanism

We will get started by reviewing the RDP and privacy profile bounds for the Gaussian mechanism. Once warmed up, we then present the RDP and privacy profile bounds for objective perturbation in Sections C and D.

Consider the Gaussian mechanism defined by $\mathcal{M}(Z) = f(Z) + \mathcal{N}(0, \sigma^2 I_d)$, for a function $f : \mathcal{Z} \to \mathbb{R}^d$ with global sensitivity $\Delta_f = \max_{Z \simeq Z'} ||f(Z) - f(Z')||_2$.

## B.1   Privacy profile of the Gaussian Mechanism

**Analytic Gaussian mechanism (Balle & Wang, 2018).**

Let $P$ and $Q$ be the density functions of $\mathcal{N}(\Delta_f, \sigma^2)$ and $\mathcal{N}(0, \sigma^2)$, respectively. Then $(P, Q)$ is a dominating pair of distributions for $\mathcal{M}$, and $\mathcal{M}$ is tightly $(\epsilon, \delta(\epsilon))$-DP for

$$\delta(\epsilon) = H_{e^\epsilon}(P||Q) = \Phi\left(-\frac{\epsilon\sigma}{\Delta_f} + \frac{\Delta_f}{2\sigma}\right) - e^\epsilon\Phi\left(-\frac{\epsilon\sigma}{\Delta_f} - \frac{\Delta_f}{2\sigma}\right), \tag{B.1}$$

where $\Phi$ denotes the CDF of the standard univariate Gaussian distribution.

We can analytically express a tight upper bound for the Gaussian mechanism above, but in general numerical methods are needed to evaluate the hockey-stick divergence for dominating pairs of distributions. This is discussed with more detail in Section D.

## B.2   RDP Analysis of the Gaussian Mechanism

*Theorem* B.1 (RDP guarantees of the Gaussian mechanism). *The Gaussian mechanism $\mathcal{M}$ satisfies $(\alpha, \epsilon)$-RDP for $\alpha > 1$ and $\epsilon = \frac{\Delta_f^2\alpha}{2\sigma^2}$.*

## C  RDP analysis of objective perturbation

In this section we present the proof of Theorem 3.2, one of our main privacy results: an RDP bound on the objective perturbation mechanism.

*Proof of Theorem 3.2.* Recall from Definition 2.7 that the objective perturbation mechanism $\hat{\theta}^P : \mathcal{Z}^* \to \mathbb{R}^d$ satisfies $\epsilon(\alpha)$-Rényi differential privacy if for all neighboring datasets $Z$ and $Z'$,

$$\mathcal{D}_\alpha \left( \hat{\theta}^P(Z) \,||\, \hat{\theta}^P(Z') \right) \le \epsilon(\alpha).$$

Assume that $Z \in \mathcal{Z}^n$ and construct $Z' \in \mathcal{Z}^{n+}$ by adding a datapoint $z$ to $Z$. Note that this convention (while convenient for writing down the PLRV of objective perturbation) comes *with* loss of generality. As a consequence of asymmetry[3], the upper bound on the RDP must satisfy

$$\max \left( D_\alpha \left( \hat{\theta}^P(Z) \,||\, \hat{\theta}^P(Z') \right), \, D_\alpha \left( \hat{\theta}^P(Z') \,||\, \hat{\theta}^P(Z) \right) \right) \le \epsilon(\alpha).$$

We will calculate the Rényi divergence $D_\alpha \left( \hat{\theta}^P(Z) \,||\, \hat{\theta}^P(Z') \right)$ of objective perturbation under a change of measure:

$$D_\alpha(P \,||\, Q) = \frac{1}{\alpha - 1} \log \mathbb{E}_{x \sim Q} \left[ \left( \frac{P(x)}{Q(x)} \right)^\alpha \right] = \frac{1}{\alpha - 1} \log \mathbb{E}_{x \sim P} \left[ \left( \frac{P(x)}{Q(x)} \right)^{\alpha - 1} \right].$$

Let $R(\theta^P) := \frac{\Pr[\hat{\theta}^P(Z) = \theta^P]}{\Pr[\hat{\theta}^P(Z') = \theta^P]}$ be shorthand for the probability density ratio at output $\theta^P$. Then

$$
\begin{aligned}
D_\alpha(\hat{\theta}^P(Z) \,||\, \hat{\theta}^P(Z')) &= \frac{1}{\alpha - 1} \log \mathbb{E}_{\theta^P \sim \hat{\theta}^P(Z)} \left[ R(\theta^P)^{(\alpha - 1)} \right] \\
&= \frac{1}{\alpha - 1} \log \mathbb{E}_{\theta^P \sim \hat{\theta}^P(Z)} \left[ e^{\log \left[ R(\theta^P)^{(\alpha - 1)} \right]} \right] \\
&= \frac{1}{\alpha - 1} \log \mathbb{E}_{\theta^P \sim \hat{\theta}^P(Z)} \left[ e^{(\alpha - 1) \log R(\theta^P)} \right] \\
&\le \frac{1}{\alpha - 1} \log \mathbb{E}_{\theta^P \sim \hat{\theta}^P(Z)} \left[ e^{(\alpha - 1) \left| \log R(\theta^P) \right|} \right].
\end{aligned}
$$

Denote $J(\theta; Z) = \sum_{z \in Z} \ell(\theta; z) + \frac{\lambda}{2} ||\theta||_2^2$ and $\mu(\theta, Z, z) = x^T \left( \nabla^2 J(\theta; Z) \right)^{-1} x$.

From Lemma K.9 and the $\lambda$-strong convexity of $J(\theta; Z)$, we can bound $\mu(\theta, Z, z)$ by $\frac{||x||_2^2}{\lambda} \le \frac{1}{\lambda}$.

We also know from the $L$-Lipschitzness of $\ell(\theta; z)$ and the $\beta$-smoothness of $f(x^T \theta; y)$ that $||\nabla \ell(\theta; z)||_2 \le L$ and $f''(x^T \theta; y) \le \beta$ for all $\theta \in \mathbb{R}^d$ and $z = (x, y) \in \mathcal{Z}$.

Then using the GLM assumption, from Redberg & Wang (2021) we can bound the absolute value of the log-probability ratio for any $\theta^P$ as

$$
\begin{aligned}
\left| \log \frac{\Pr \left[ \hat{\theta}^P(Z) = \theta^P \right]}{\Pr \left[ \hat{\theta}^P(Z') = \theta^P \right]} \right| &\le \left| -\log \left( 1 - f''(x^T \theta; y) \mu(\theta^P, Z, z) \right) - \frac{1}{2\sigma^2} ||\nabla \ell(\theta^P; z)||_2^2 - \frac{1}{\sigma^2} \nabla J(\theta^P; Z)^T \nabla \ell(\theta; z) \right| \\
&\le \left| -\log \left( 1 - f''(x^T \theta^P; y) \mu(\theta^P, Z, z) \right) \right| + \frac{1}{2\sigma^2} ||\nabla \ell(\theta^P; z)||_2^2 + \frac{1}{\sigma^2} \left| \nabla J(\theta^P; Z)^T \nabla \ell(\theta^P; z) \right| \\
&\le \left| -\log \left( 1 - \frac{\beta}{\lambda} \right) \right| + \frac{L^2}{2\sigma^2} + \frac{1}{\sigma^2} \left| \nabla J(\theta^P; Z)^T \nabla \ell(\theta^P; z) \right|.
\end{aligned}
$$

It is more challenging to find a data-independent bound for the third term due to the the shared dependence on $\theta^P$.

---

[3]This is in contrast to the symmetry of the Gaussian mechanism $\mathcal{M}(Z) = f(Z) + \mathcal{N}(0, \sigma^2 I_d)$, in which case $\epsilon(\alpha)$ can be calculated exactly as $D_\alpha \left( \mathcal{N}(0, \sigma^2 I_d) \,||\, \mathcal{N}(\Delta_f, \sigma^2 I_d) \right) = \frac{\alpha \Delta_f^2}{2\sigma^2} = D_\alpha \left( \mathcal{N}(\Delta_f, \sigma^2 I_d) \,||\, \mathcal{N}(0, \sigma^2 I_d) \right)$.

Recall that $b \sim \mathcal{N}(0, \sigma^2 I_d)$ is the noise vector in the perturbed objective. By first-order conditions at the minimizer $\theta^P$,

$$b = -\nabla J(\theta^P; Z).$$

If $\theta^P$ were fixed (or if $\theta^P$ were independent to $b$), the quantity $\nabla J(\theta^P; Z)^T \nabla \ell(\theta^P; z) = -b^T \nabla \ell(\theta^P; z)$ would have been distributed as a univariate Gaussian $\mathcal{N}(0, \sigma^2 \|\nabla \ell(\theta^P; z)\|_2^2)$. Unfortunately in our case $\theta^P$ is a random variable, and consequently we don't have the tools to understand the distribution of $\nabla J(\theta^P; Z)^T \nabla \ell(\theta^P; z)$ for an arbitrary loss function.

But using the GLM assumption on the loss function, we can write

$$\nabla J(\theta; Z)^T \nabla \ell(\theta; z) = -b^T x f'(x^T \theta, y).$$

Observe that $x$ is fixed w.r.t. $b$ so that $-b^T x \sim \mathcal{N}(0, \sigma^2 \|x\|_2^2)$, and $f'(x^T \theta, y)$ is a scalar. So while this scalar is a random variable that still depends on $b$ in a complicated way, the worst possible dependence can be more easily quantified without incurring additional dimension dependence.

By the $L$-Lipschitz assumption and $|ab| \le |a||b|$, we obtain the following bound:

$$
\begin{aligned}
\left| \nabla J(\theta^P; Z)^T \nabla \ell(\theta^P; z) \right| &= \left| f'(x^T \theta^P; y) \nabla J(\theta^P; Z)^T x \right| \\
&\le L \left| \nabla J(\theta^P; Z)^T x \right|.
\end{aligned}
$$

This is much better! By first-order conditions we can then see

$$L \left| \nabla J(\theta^P; Z)^T x \right| = L \left| b^T x \right| = \left| \mathcal{N}(0, \sigma^2 \|x\|^2 L^2) \right| \sim \text{Half-Normal}(\sigma L \|x\|).$$

Now we can bound

$$
\begin{aligned}
\left| \log R(\theta^P) \right| &\le \left| -\log\left(1 - \frac{\beta}{\lambda}\right) \right| + \frac{L^2}{2\sigma^2} + \frac{1}{\sigma^2} \left| \nabla J(\theta^P; Z)^T \nabla \ell(\theta^P; z) \right| \\
&\le \left| -\log\left(1 - \frac{\beta}{\lambda}\right) \right| + \frac{L^2}{2\sigma^2} + \frac{L}{\sigma^2} \left| \nabla J(\theta^P; Z)^T x \right|.
\end{aligned}
$$

Plugging in this bound on $\left| \log R(\theta^P) \right|$,

$$
\begin{aligned}
D_\alpha(\hat{\theta}^P(Z) \,\|\, \hat{\theta}^P(Z')) &\le \frac{1}{\alpha - 1} \log \mathbb{E}_{\theta^P \sim \hat{\theta}^P(Z)} \left[ e^{(\alpha - 1)\left| \log R(\theta^P) \right|} \right] \\
&\le \frac{1}{\alpha - 1} \log \mathbb{E}_{\theta^P \sim \hat{\theta}^P(Z)} \left[ e^{(\alpha - 1)\left[ \left| -\log(1 - \frac{\beta}{\lambda}) \right| + \frac{L^2}{2\sigma^2} \right]} e^{(\alpha - 1) \cdot \frac{L}{\sigma^2} \left| \nabla J(\theta^P; Z)^T x \right|} \right] \\
&= \left| -\log\left(1 - \frac{\beta}{\lambda}\right) \right| + \frac{L}{2\sigma^2} + \frac{1}{\alpha - 1} \log \mathbb{E}_{\theta^P \sim \hat{\theta}^P(Z)} \left[ e^{(\alpha - 1) \cdot \frac{L}{\sigma^2} \left| \nabla J(\theta^P; Z)^T x \right|} \right].
\end{aligned}
$$

Let $p_\sigma$ be the probability density function of $b \sim \mathcal{N}(0, \sigma^2 I_d)$, and $p_\Theta$ the probability density function of $\theta^P \sim \hat{\theta}^P(Z)$. We know from Lemmas K.6 and K.7 that $\partial \theta = \left| \det \frac{\partial \theta}{\partial b} \right| \partial b$, and

$$p_\Theta(\theta) = \left| \det \frac{\partial b}{\partial \theta} \right| p_\sigma(b).$$

We also know from Lemma K.8 that $\left| \det \frac{\partial \theta}{\partial b} \right| \cdot \left| \det \frac{\partial b}{\partial \theta} \right| = 1$.

Using the change of variables $b = -\nabla J(\theta^P; Z)$ and $b^T x = u \sim \mathcal{N}(0, \sigma^2 ||x||_2^2)$, we have

$$\mathbb{E}_{\theta^P \sim \hat{\theta}^P(Z)} \left[ e^{(\alpha-1) \cdot \frac{L}{\sigma^2} |\nabla J(\theta^P; Z)^T x|} \right] = \int_{\mathbb{R}^d} e^{(\alpha-1) \cdot \frac{L}{\sigma^2} |\nabla J(\theta^P; Z)^T x|} p_\Theta \left( \theta^P \right) \partial \theta$$

$$= \int_{\mathbb{R}^d} e^{(\alpha-1) \cdot \frac{L}{\sigma^2} |b^T x|} \left| \det \frac{\partial b}{\partial \theta} \right| p_\sigma(b) \left| \det \frac{\partial \theta}{\partial b} \right| \partial b$$

$$= \int_{\mathbb{R}^d} e^{(\alpha-1) \cdot \frac{L}{\sigma^2} |b^T x|} p_\sigma(b) \partial b$$

$$= \mathbb{E}_{b \sim \mathcal{N}(0, \sigma^2 I_d)} \left[ e^{(\alpha-1) \cdot \frac{L}{\sigma^2} |b^T x|} \right]$$

$$= \mathbb{E}_{u \sim \mathcal{N}(0, \sigma^2 ||x||_2^2)} \left[ e^{(\alpha-1) |\frac{L}{\sigma^2} u^2|} \right]$$

$$\leq \mathbb{E}_{\zeta \sim \mathcal{N}\left(0, \frac{L^2}{\sigma^2}\right)} \left[ e^{(\alpha-1)|\zeta|} \right].$$

In the last line, we applied our assumption that $||x|| \leq 1$ and the fact that the MGF of a half-normal R.V. increases monotonically when its scale parameter gets larger.

The above bound holds for the reverse Rényi divergence $D_\alpha(\hat{\theta}^P(Z') \,||\, \hat{\theta}^P(Z))$. Observe that

$$D_\alpha(\hat{\theta}^P(Z') \,||\, \hat{\theta}^P(Z)) \leq \frac{1}{\alpha - 1} \log \mathbb{E}_{\theta^P \sim \hat{\theta}^P(Z')} \left[ e^{(\alpha-1)|\log R(\theta^P)|} \right].$$

This is because $\log \frac{\Pr[\hat{\theta}^P(Z') = \theta^P]}{\Pr[\hat{\theta}^P(Z) = \theta^P]} = -\log R(\theta^P) \leq |\log R(\theta^P)|$. If we use the change of variables $b = -\nabla J(\theta^P; Z')$ for the reverse direction, the above calculation works out identically (the difference is that $p_\Theta$ and the bijection between $b$ and $\theta^P$ are different under $Z$ and $Z'$ — but the determinant of the mapping cancels out with its inverse just the same).

We've shown $\max \left( D_\alpha\left(\hat{\theta}^P(Z) \,||\, \hat{\theta}^P(Z')\right), D_\alpha\left(\hat{\theta}^P(Z') \,||\, \hat{\theta}^P(Z)\right) \right) \leq \epsilon(\alpha)$ for any neighboring datasets $Z$ and $Z'$, where

$$\epsilon(\alpha) = -\log \left( 1 - \frac{\beta}{\lambda} \right) + \frac{L}{2\sigma^2} + \mathbb{E} \left[ e^{(\alpha-1)\left|\mathcal{N}\left(0, \frac{L^2}{\sigma^2}\right)\right|} \right].$$

$\square$

## C.1 Linearized RDP Bound for Objective Perturbation

In our calculation of the RDP for objective perturbation, we needed to take an absolute value of the privacy loss random variable in order to handle negative values. But in doing so we end up with a quantity that depends on the moments of the *half-normal* distribution rather than those of the normal distribution, which gives us a looser bound. Can we avoid having to make this compromise? In this section we demonstrate that a linearization of the first-order conditions on the perturbed and unperturbed objective functions provides a more precise analysis of the PLRV of objective perturbation, translating to a tighter RDP bound in some regimes.

Recall that the objective perturbation mechanism is given by

$$\hat{\theta}^P(Z) = \sum_{i=1}^n \ell(\theta; z_i) + \frac{\lambda}{2} ||\theta||_2^2 + b^T \theta, \tag{C.1}$$

where $b \sim \mathcal{N}(0, \sigma^2 I_d)$.

From the non-linearized RDP calculation, we know that for any neighboring datasets $Z$ and $Z'$,

$$D_\alpha \left( \hat{\theta}^P(Z) \,||\, \hat{\theta}^P(Z') \right) \leq \left| -\log \left( 1 - \frac{\beta}{\lambda} \right) \right| + \frac{L^2}{2\sigma^2} + \frac{1}{\alpha - 1} \log \mathbb{E}_{b \sim \mathcal{N}(0, \sigma^2 I_d)} \left[ e^{(\alpha-1)b^T \nabla \ell(\theta^P)} \right],$$

where $\theta^P$ is the output of the objective perturbation mechanism given the noise vector $b$. We can write

$$b^T \nabla \ell(\theta^P) = b^T \nabla \ell(\theta_\lambda^*) + b^T \left[ \nabla \ell(\theta^P) - \nabla \ell(\theta_\lambda^*) \right].$$

The first term $t_1 = b^T \nabla \ell(\theta_\lambda^*)$ is a univariate Gaussian $t_1 \sim \mathcal{N}(0, \sigma^2 ||\nabla \ell(\theta_\lambda^*)||_2^2)$ because $\theta_\lambda^*$ is fixed w.r.t. $b$. We can bound the second term $t_2 = b^T \left[ \nabla \ell(\theta^P) - \nabla \ell(\theta_\lambda^*) \right]$ using our assumptions on the loss function $\ell(\theta)$.

By assumption, the loss function has GLM structure $\ell(\theta; z) = f(x^T \theta; y)$. We can therefore write

$$b^T \left[ \nabla \ell(\theta^P) - \nabla \ell(\theta_\lambda^*) \right] = b^T \left[ f'(x^T \theta^P; y)x - f'(x^T \theta_\lambda^*; y)x \right]$$
$$= b^T x \left( f'(x^T \theta^P; y) - f'(x^T \theta_\lambda^*; y) \right)$$

We have furthermore assumed that the function $f$ is $\beta$-smooth, so that for any $z = (x, y)$ and $\theta^P, \theta_\lambda^* \in \mathbb{R}^d$ we have

$$\left| f'(x^T \theta^P; y) - f'(x^T \theta_\lambda^*; y) \right| \le \beta \left| x^T \theta^P - x^T \theta_\lambda^* \right|$$
$$= \beta \left| x^T \left[ \theta^P - \theta_\lambda^* \right] \right|.$$

We will next apply Taylor's Theorem to rewrite $\theta^P - \theta_\lambda^*$.

Recall that $\theta^P$ is the minimizer of the perturbed objective:

$$\theta^P = \arg\min \left( \sum_{i=1}^{n} \ell(\theta; z_i) + \frac{\lambda}{2} ||\theta||_2^2 + b^T \theta \right); \tag{C.2}$$

and $\theta_\lambda^*$ is the minimizer of the (non-private) regularized objective:

$$\theta_\lambda^* = \arg\min \left( \sum_{i=1}^{n} \ell(\theta; z_i) + \frac{\lambda}{2} ||\theta||_2^2 \right). \tag{C.3}$$

Parameterize the line segment between $\theta^P$ and $\theta_\lambda^*$ by $t \in [0, 1]$, i.e. the line segment is $t(\theta^P - \theta_\lambda^*) + \theta^P$. By Taylor's Theorem, there exists $\theta' = t'(\theta^P - \theta_\lambda^*) + \theta^P$ for some $t' \in [0, 1]$ such that

$$\nabla \ell(\theta^P) - \nabla \ell(\theta_\lambda^*) = \nabla^2 \ell(\theta')(\theta^P - \theta_\lambda^*).$$

By first-order conditions on Equations C.2 and C.3,

$$\nabla \mathcal{L}(\theta^P) + \lambda \theta^P + b = 0; \tag{C.4}$$
$$\nabla \mathcal{L}(\theta_\lambda^*) + \lambda \theta_\lambda^* = 0. \tag{C.5}$$

Then subtracting Equation C.5 from Equation C.4, we have that

$$\nabla \mathcal{L}(\theta^P) - \nabla \mathcal{L}(\theta_\lambda^*) + \lambda(\theta^P - \theta_\lambda^*) + b = 0. \tag{C.6}$$

Again applying Taylor's theorem, there exists $\theta'' = t''(\theta^P - \theta_\lambda^*) + \theta^P$ for some $t'' \in [0, 1)$) such that

$$\nabla \mathcal{L}(\theta^P) - \nabla \mathcal{L}(\theta_\lambda^*) = \nabla^2 \mathcal{L}(\theta'')(\theta^P - \theta_\lambda^*). \tag{C.7}$$

Putting together Equations C.6 and C.7 we then have

$$\theta^P - \theta_\lambda^* = - \left( \nabla^2 \mathcal{L}(\theta'') + \lambda I_d \right)^{-1} b. \tag{C.8}$$

So we now have

$$b^T \left[ \nabla \ell(\theta^P) - \nabla \ell(\theta_\lambda^*) \right] \le \beta \left| b^T x \right| \left| x^T \left( \theta^P - \theta_\lambda^* \right) \right|$$
$$\le \beta \left| b^T x \right| \left| x^T \left( \nabla^2 \mathcal{L}(\theta_\lambda^*) + \lambda I_d \right)^{-1} b \right|. \tag{C.9}$$

Note that since $e^x > 0$ for all $x \in \mathbb{R}$, we have that $\mathbb{E} \left[ |e^x| \right] = \mathbb{E} \left[ e^x \right]$.

Let $a := \frac{1}{\sigma^2} b^T \nabla \ell(\theta^*)$ and $c := \frac{1}{\sigma^2} b^T \left[ \nabla \ell(\theta^P) - \nabla \ell(\theta^*) \right]$. Then by Holder's inequality,

$$\mathbb{E}\left[e^{(\alpha-1)a}e^{(\alpha-1)c}\right] = \mathbb{E}\left[\left|e^{(\alpha-1)a}e^{(\alpha-1)c}\right|\right]$$

$$\leq \mathbb{E}\left[\left|e^{(\alpha-1)a}\right|^{p}\right]^{\frac{1}{p}}\mathbb{E}\left[\left|e^{(\alpha-1)c}\right|^{q}\right]^{\frac{1}{q}}$$

$$= \mathbb{E}\left[e^{(p\alpha-p)a}\right]^{\frac{1}{p}}\mathbb{E}\left[e^{(q\alpha-q)c}\right]^{\frac{1}{q}}.$$

By the GLM assumption, $b^{T}\nabla\ell(\theta_{\lambda}^{*};z) = f'(x^{T}\theta_{\lambda}^{*};y)b^{T}x$. Then

$$\mathbb{E}_{b\sim\mathcal{N}(0,\sigma^2 I_d)}\left[e^{(p\alpha-p)\frac{1}{\sigma^2}b^{T}\nabla\ell(\theta^{P})}\right] = \mathbb{E}_{b\sim\mathcal{N}(0,\sigma^2 I_d)}\left[e^{(p\alpha-p)\frac{1}{\sigma^2}f'(x^{T}\theta_{\lambda}^{*};y)b^{T}x}\right]$$

$$= \mathbb{E}_{u_1\sim\mathcal{N}\left(0,f'(x^{T}\theta_{\lambda}^{*};y)^2\|x\|_2^2\frac{1}{\sigma^2}\right)}\left[e^{(p\alpha-p)u_1}\right]$$

$$\leq \mathbb{E}_{u_2\sim\mathcal{N}\left(0,\frac{L^2}{\sigma^2}\right)}\left[e^{(p\alpha-p)u_2}\right].$$

Above, we've applied the assumption that $\|x\|_2 \leq 1$ and the fact that the MGF of a normal R.V. increases monotonically when its scale parameter gets larger (Lemma K.11). By Lemma K.9, we have that $x^{T}\left(\nabla^2\mathcal{L}(\theta_{\lambda}^{*}) + I_d\right)^{-2}x \leq \frac{\|x\|_2^2}{\lambda^2}$. From C.9 we also have

$$\mathbb{E}_{b\sim\mathcal{N}(0,\sigma^2 I_d)}\left[e^{(q\alpha-q)\frac{1}{\sigma^2}b^{T}\left[\nabla\ell(\theta^{P})-\nabla\ell(\theta_{\lambda}^{*})\right]}\right] \leq \mathbb{E}_{b\sim\mathcal{N}(0,\sigma^2 I_d)}\left[e^{(q\alpha-q)\beta\left|b^{T}x\right|\left|x^{T}\left(\nabla^2\mathcal{L}(\theta^{*})+\lambda I_d\right)^{-1}b\right|}\right]$$

Define $z_1 := b^{T}x$ and $z_2 := x^{T}\left(\nabla^2\mathcal{L}(\theta_{\lambda}^{*}) + \lambda I_d\right)^{-1}b$, and observe

$$z_1 \sim \mathcal{N}\left(0,\sigma^2\|x\|_2^2\right),$$

$$z_2 \sim \mathcal{N}\left(0,\sigma^2 x^{T}\left(\nabla^2\mathcal{L}(\theta_{\lambda}^{*})+I_d\right)^{-2}x\right).$$

Note that our approach below is agnostic to the relationship between $|z_1|$ and $|z_2|$; in reality, they depend on each other through the noise vector $b$. Again applying the assumption $\|x\|_2^2 \leq 1$ and Lemma K.11 (while not forgetting that the random variables $z_1$ and $z_2$ depend on each other through $b$), we get

$$\mathbb{E}_{b\sim\mathcal{N}(0,\sigma^2 I_d)}\left[e^{(q\alpha-q)\beta\left|b^{T}x\right|\left|x^{T}\left(\nabla^2\mathcal{L}(\theta_{\lambda}^{*})+\lambda I_d\right)^{-1}b\right|}\right] = \mathbb{E}_{z_1,z_2}\left[e^{(q\alpha-q)\beta|z_1||z_2|}\right]$$

$$\leq \mathbb{E}_{z_3\sim\mathcal{N}(0,\sigma^2),z_4\sim\mathcal{N}(0,\frac{\sigma^2}{\lambda^2})}\left[e^{(q\alpha-q)\beta|z_3||z_4|}\right]$$

$$= \mathbb{E}_{z_3\sim\mathcal{N}(0,\sigma^2),z_5\sim\mathcal{N}(0,\sigma^2)}\left[e^{(q\alpha-q)\frac{\beta}{\lambda}|z_3||z_5|}\right]$$

$$\leq \mathbb{E}_{z\sim\mathcal{N}(0,\sigma^2)}\left[e^{(q\alpha-q)\frac{\beta}{\lambda}z^2}\right].$$

So altogether, for $p,q$ such that $\frac{1}{p} + \frac{1}{q} = 1$, we get

$$D_{\alpha}\left(\hat{\theta}^{P}(Z)\,\|\,\hat{\theta}^{P}(Z')\right) \leq$$

$$-\log\left(1-\frac{\beta}{\lambda}\right) + \frac{L^2}{2\sigma^2} + \frac{1}{\alpha-1}\log\mathbb{E}_{u\sim\mathcal{N}(0,\frac{L^2}{\sigma^2})}\left[e^{(p\alpha-p)u}\right]^{\frac{1}{p}}\mathbb{E}_{z\sim\mathcal{N}(0,\sigma^2)}\left[e^{(q\alpha-q)\frac{\beta}{\lambda}z^2}\right]^{\frac{1}{q}}.$$

# D  Hockey-stick Divergence Analysis of Objective Perturbation

## D.1  Further Details on Hockey-stick Divergence Analysis

Using dominating pairs of distributions (Def. 2.5) for all the individual mechanisms in an adaptive composition, we can obtain accurate $(\epsilon,\delta)$-bounds for the whole composition. For this end we need the following result.

*Theorem* D.1 (Zhu et al. 2022). *If* $(P, Q)$ *dominates* $\mathcal{M}$ *and* $(P', Q')$ *dominates* $\mathcal{M}'$ *for all inputs of* $\mathcal{M}'$, *then* $(P \times P', Q \times Q')$ *dominates the adaptive composition* $\mathcal{M} \circ \mathcal{M}'$.

To get the hockey-stick divergence from $P \times P'$ to $Q \times Q'$ into an efficiently computable form, we express it using so called privacy loss random variables (recall Def. K.4). If $P$ and $Q$ are probability density functions, the privacy loss function $\mathcal{L}_{P/Q}$ is defined as

$$\mathcal{L}_{P/Q}(x) = \log \frac{P(x)}{Q(x)}$$

and the privacy loss random variable (PLRV) $\omega_{P/Q}$ as

$$\omega_{P/Q} = \mathcal{L}_{P/Q}(x), \quad x \sim P(x).$$

The $\delta(\epsilon)$-bounds can be represented using the following representation that involves the PLRV.

*Theorem* D.2 (Gopi et al. 2021). *We have:*

$$H_{e^\epsilon}(P||Q) = \underset{x \sim P}{\mathbb{E}} \left[ 1 - e^{\epsilon - \mathcal{L}_{P/Q}(x)} \right]_+ = \underset{s \sim \omega_{P/Q}}{\mathbb{E}} \left[ 1 - e^{\epsilon - s} \right]_+. \tag{D.1}$$

*Moreover, if* $\omega_{P/Q}$ *is the PLRV for the pair of distributions* $(P, Q)$ *and* $\omega_{P'/Q'}$ *the PLRV for the pair of distributions* $(P', Q')$, *then the PLRV for the pair of distributions* $(P \times P', Q \times Q')$ *is given by* $\omega_{P/Q} + \omega_{P'/Q'}$.

By Theorem D.2, to computing accurate $(\epsilon, \delta)$-bounds for compositions, it suffices that we can evaluate integrals of the form $\mathbb{E}_{s \sim \omega_1 + \ldots + \omega_k} \left[ 1 - e^{\epsilon - s} \right]_+$. For this we can use the Fast Fourier Transform (FFT)-based method by Koskela et al. (2021), where the distribution of each PLRV is truncated and placed on an equidistant numerical grid over an interval $[-L, L]$, where $L > 0$ is a pre-defined parameter. The distributions for the sums of the PLRVs are given by convolutions of the individual distributions and can be evaluated using the FFT algorithm. By a careful error analysis the error incurred by the numerical method can be bounded and an upper $\delta(\epsilon)$-bound obtained. For accurately carrying out this numerical computation one could also use, for example, the FFT-based method proposed by Gopi et al. (2021).

### D.2  Proof of Theorem 3.1

Before giving a proof to Thm. 3.1, we first give the following bound which is a hockey-stick equivalent of the moment-generating function bound given in Thm. 3.2.

*Lemma* D.3. *Let* $\epsilon \in \mathbb{R}$ *and let the objective perturbation mechanism* $\hat{\theta}^P$ *be defined as in Section 2.2. Let* $||\nabla \ell(\theta; z)||_2 \leq L$ *and* $\nabla^2 \ell(\theta; z) \prec \beta I_d$ *for all* $\theta \in \Theta$ *and* $z \in \mathcal{X} \times \mathcal{Y}$. *Then, for any neighboring datasets* $Z$ *and* $Z'$, *we have:*

$$H_{e^\epsilon}\left(\hat{\theta}^P(Z) || \hat{\theta}^P(Z')\right) \leq \mathbf{E}_{s \sim \omega}[1 - e^{\epsilon - s}]_+, \tag{D.2}$$

*where* $\omega \sim \left| \log \left( 1 - \frac{\beta}{\lambda} \right) \right| + \frac{L^2}{2\sigma^2} + \left| \mathcal{N} \left( \frac{||x||^2 L^2}{\sigma^2} \right) \right|$.

*Proof.* The proof goes analogously to the proof of Thm. 3.2. Let $Z$ and $Z'$ be any neighboring datasets. Following the proof of Thm. 3.2, denote the privacy loss

$$R(\theta) := \frac{\Pr\left[\hat{\theta}^P(Z) = \theta\right]}{\Pr\left[\hat{\theta}^P(Z') = \theta\right]}.$$

By Thm. D.2 and by using the reasoning of the proof of Thm. 3.2 for the moment-generating function, we have

$$\begin{aligned} H_{e^\epsilon}\left(\hat{\theta}^P(Z) || \hat{\theta}^P(Z')\right) &= \mathbf{E}_{\theta \sim \hat{\theta}^P(Z)}[1 - e^{\epsilon - \log R(\theta)}]_+ \\ &\leq \mathbf{E}_{\theta \sim \hat{\theta}^P(Z)}[1 - e^{\epsilon - |\log R(\theta)|}]_+ \\ &\leq \mathbf{E}_{s \sim \left| \mathcal{N}\left(\frac{||x||^2 L^2}{\sigma^2}\right)\right|}[1 - e^{\epsilon - \left|\log\left(1 - \frac{\beta}{\lambda}\right)\right| - \frac{L^2}{2\sigma^2} - s}]_+, \end{aligned} \tag{D.3}$$

where the inequalities follow from the fact that the function $f(s) = [1 - e^{\epsilon - s}]_+$ is monotonically increasing function w.r.t. $s$ for all $\epsilon \in \mathbb{R}$ and from the bound for $|R(\theta)|$ used in the proof of Thm. 3.2. $\square$

*Proof of Theorem 3.1.* We use Lemma D.3 and simply upper bound the right-hand side of the inequality (D.2).

We first show that if $\|x\| \leq 1$, then for all $\epsilon \in \mathbb{R}$

$$\mathbf{E}_{s \sim \left|\log\left(1-\frac{\beta}{\lambda}\right)\right|+\frac{L^2}{2\sigma^2}\left|\mathcal{N}\left(0,\frac{\|x\|^2 L^2}{\sigma^2}\right)\right|}\left[1-e^{\epsilon-s}\right]_+ \leq \mathbf{E}_{s \sim \left|\log\left(1-\frac{\beta}{\lambda}\right)\right|+\frac{L^2}{2\sigma^2}\left|\mathcal{N}\left(0,\frac{L^2}{\sigma^2}\right)\right|}\left[1-e^{\epsilon-s}\right]_+. \quad \text{(D.4)}$$

Denote $\hat{\epsilon} = \epsilon - \left|\log\left(1-\frac{\beta}{\lambda}\right)\right| - \frac{L^2}{2\sigma^2}$. Consider first the case $\hat{\epsilon} \geq 0$. Then, we have:

$$\mathbf{E}_{s \sim \left|\mathcal{N}\left(0,\frac{\|x\|^2 L^2}{\sigma^2}\right)\right|}\left[1-e^{\epsilon-\left|\log\left(1-\frac{\beta}{\lambda}\right)\right|-\frac{L^2}{2\sigma^2}-s}\right]_+ = 2 \cdot \mathbf{E}_{s \sim \mathcal{N}\left(0,\frac{\|x\|^2 L^2}{\sigma^2}\right)}\left[1-e^{\hat{\epsilon}-s}\right]_+$$

$$\leq 2 \cdot \mathbf{E}_{s \sim \mathcal{N}\left(0,\frac{L^2}{\sigma^2}\right)}\left[1-e^{\hat{\epsilon}-s}\right]_+$$

$$= \mathbf{E}_{s \sim \left|\mathcal{N}\left(0,\frac{L^2}{\sigma^2}\right)\right|}\left[1-e^{\epsilon-\left|\log\left(1-\frac{\beta}{\lambda}\right)\right|-\frac{L^2}{2\sigma^2}-s}\right]_+,$$

$$\text{(D.5)}$$

where the first equality follows from the fact that for $s \geq \hat{\epsilon}$, the density function of the half-normal random variable is positive and 2 times the density of the corresponding normal distribution. The inequality follows from Lemma D.8, as

$$\mathbf{E}_{s \sim \mathcal{N}\left(0,\frac{\|x\|^2 L^2}{\sigma^2}\right)}\left[1-e^{\hat{\epsilon}-s}\right]_+ = \int_{\hat{\epsilon}}^{\infty} f_{0,\frac{\|x\|^2 L^2}{\sigma^2}}(x)(1-e^{\hat{\epsilon}-x})\,\mathrm{d}x.$$

Next, consider the case $\hat{\epsilon} < 0$. Then:

$$\mathbf{E}_{s \sim \left|\mathcal{N}\left(0,\frac{\|x\|^2 L^2}{\sigma^2}\right)\right|}\left[1-e^{\epsilon-\left|\log\left(1-\frac{\beta}{\lambda}\right)\right|-\frac{L^2}{2\sigma^2}-s}\right]_+ = 2 \cdot \int_0^{\infty} f_{0,\frac{\|x\|^2 L^2}{\sigma^2}}(x)(1-e^{\epsilon-\left|\log\left(1-\frac{\beta}{\lambda}\right)\right|-\frac{L^2}{2\sigma^2}-x})\,\mathrm{d}x$$

$$\leq 2 \cdot \int_0^{\infty} f_{0,\frac{L^2}{\sigma^2}}(x)(1-e^{\epsilon-\left|\log\left(1-\frac{\beta}{\lambda}\right)\right|-\frac{L^2}{2\sigma^2}-x})\,\mathrm{d}x$$

$$= \mathbf{E}_{s \sim \left|\mathcal{N}\left(0,\frac{L^2}{\sigma^2}\right)\right|}\left[1-e^{\epsilon-\left|\log\left(1-\frac{\beta}{\lambda}\right)\right|-\frac{L^2}{2\sigma^2}-s}\right]_+.$$

$$\text{(D.6)}$$

where the inequality follows from Lemma D.8. Inequalities (D.5) and (D.6) together give (D.4).

Then, we show that for all $\epsilon \in \mathbb{R}$,

$$\mathbf{E}_{s \sim \left|\log\left(1-\frac{\beta}{\lambda}\right)\right|+\frac{L^2}{2\sigma^2}\left|\mathcal{N}\left(0,\frac{L^2}{\sigma^2}\right)\right|}\left[1-e^{\epsilon-s}\right]_+ = \begin{cases} 2 \cdot H_{e^{\tilde{\epsilon}}}\left(P||Q\right), & \text{if } \hat{\epsilon} \geq 0, \\ (1-e^{\hat{\epsilon}}) + e^{\hat{\epsilon}} \cdot 2 \cdot H_{e^{\frac{L^2}{\sigma^2}}}\left(P||Q\right), & \text{otherwise.} \end{cases}$$

Continuing from (D.5), by change of variables, we see that for $\hat{\epsilon} \geq 0$,

$$2 \cdot \mathbf{E}_{s \sim \mathcal{N}\left(0,\frac{L^2}{\sigma^2}\right)}\left[1-e^{\epsilon-\left|\log\left(1-\frac{\beta}{\lambda}\right)\right|-\frac{L^2}{2\sigma^2}-s}\right]_+ = 2 \cdot \mathbf{E}_{s \sim \mathcal{N}\left(\frac{L^2}{2\sigma^2},\frac{L^2}{\sigma^2}\right)}\left[1-e^{\epsilon-\left|\log\left(1-\frac{\beta}{\lambda}\right)\right|-s}\right]_+$$

$$= 2 \cdot H_{e^{\tilde{\epsilon}}}\left(P||Q\right),$$

where $\tilde{\epsilon} = \epsilon - \left|\log\left(1-\frac{\beta}{\lambda}\right)\right|$, $P$ is the density function of $\mathcal{N}(L, \sigma^2)$ and $Q$ the density function of $\mathcal{N}(0, \sigma^2)$. This follows from the fact that the PLRV determined by the pair $(P, Q)$ is distributed as $\mathcal{N}(\frac{L^2}{2\sigma^2}, \frac{L^2}{\sigma^2})$.

Continuing from (D.6), by change of variables (used after the third equality sign), we see that for $\hat{\epsilon} z 0$,

$$2 \cdot \int_0^{\infty} f_{0,\frac{L^2}{\sigma^2}}(x)(1-e^{\epsilon-\left|\log\left(1-\frac{\beta}{\lambda}\right)\right|-\frac{L^2}{2\sigma^2}-x})\,\mathrm{d}x$$

$$= 2 \cdot \int_0^{\infty} f_{0,\frac{L^2}{\sigma^2}}(x)\,\mathrm{d}x - 2 \cdot \int_0^{\infty} f_{0,\frac{L^2}{\sigma^2}}(x)(1-e^{\hat{\epsilon}-x})\,\mathrm{d}x$$

$$= (1-e^{\hat{\epsilon}}) \cdot 2 \cdot \int_0^{\infty} f_{0,\frac{L^2}{\sigma^2}}(x)\,\mathrm{d}x + e^{\hat{\epsilon}} \cdot 2 \cdot \int_0^{\infty} f_{0,\frac{L^2}{\sigma^2}}(x)(1-e^{-x})\,\mathrm{d}x$$

$$= (1-e^{\hat{\epsilon}}) + e^{\hat{\epsilon}} \cdot 2 \cdot \int_{\frac{L^2}{2\sigma^2}}^{\infty} f_{\frac{L^2}{2\sigma^2},\frac{L^2}{\sigma^2}}(x)(1-e^{\frac{L^2}{2\sigma^2}-x})\,\mathrm{d}x$$

$$= (1-e^{\hat{\epsilon}}) + e^{\hat{\epsilon}} \cdot 2 \cdot \mathbf{E}_{s \sim \mathcal{N}\left(\frac{L^2}{2\sigma^2},\frac{L^2}{\sigma^2}\right)}\left[1-e^{\frac{L^2}{2\sigma^2}-s}\right]_+$$

$$= (1-e^{\hat{\epsilon}}) + e^{\hat{\epsilon}} \cdot 2 \cdot H_{e^{\frac{L^2}{2\sigma^2}}}\left(P||Q\right),$$

where we again use the fact that the PLRV of the Gaussian mechanism with sensitivity $L$ and noise scale $\sigma$ is distributed as $\mathcal{N}(\frac{L^2}{2\sigma^2}, \frac{L^2}{\sigma^2})$. $\square$

## D.3 Dominating Pairs of Distributions for the Objective Perturbation Mechanism

From Lemma D.3 and the inequality (D.4) we have that for all $\epsilon \in \mathbb{R}$

$$H_{e^\epsilon}\left(\hat{\theta}^P(Z) \| \hat{\theta}^P(Z')\right) \leq \mathbf{E}_{\omega \sim \left|\log\left(1-\frac{\beta}{\lambda}\right)\right| + \frac{L^2}{2\sigma^2} + \left|\mathcal{N}\left(0, \frac{L^2}{\sigma^2}\right)\right|}[1 - e^{\epsilon - \omega}]_+ .$$

Thus, if we have distributions $P$ and $Q$ such that for all $\epsilon \in \mathbb{R}$

$$H_{e^\epsilon}\left(P \| Q\right) = \mathbf{E}_{\omega \sim \left|\log\left(1-\frac{\beta}{\lambda}\right)\right| + \frac{L^2}{2\sigma^2} + \left|\mathcal{N}\left(0, \frac{L^2}{\sigma^2}\right)\right|}[1 - e^{\epsilon - \omega}]_+ ,$$

then the pair $(P, Q)$ is a dominating pair of distributions for the objective perturbation mechanism. Then, by Theorem D.2, we can use this distribution $\omega$ also to compute $(\epsilon, \delta)$-bounds for compositions involving the objective perturbation mechanism. We give such a pair of distribution $(P, Q)$ explicitly in Lemma D.6 below.

In the following, we denote the density of a discrete probability mass by a Dirac delta function and use the indicator function for the continuous part of the density. The following result is a straightforward calculation.

*Lemma D.4. Let $\sigma > 0$. Let $P$ be the density function of $\left|\mathcal{N}\left(0, \sigma^2\right)\right|$, i.e.,*

$$P(x) = \frac{2}{\sqrt{2\pi\sigma^2}} e^{\frac{-x^2}{2\sigma^2}} \mathbf{1}_{[0,\infty)]}(x), \tag{D.7}$$

*where $\mathbf{1}_A(x)$ denotes the indicator function, i.e., $\mathbf{1}_{[0,\infty)]}(x) = 1$ if $x \geq 0$, else $\mathbf{1}_{[0,\infty)]}(x) = 0$. Let $L > 0$ and let $Q$ be a density function, where part of the mass of $P$ is shifted to $-\infty$:*

$$Q(x) = Q(-\infty) \cdot \delta_{-\infty}(x) + \frac{2}{\sqrt{2\pi\sigma^2}} e^{\frac{-(t+L)^2}{2\sigma^2}} \mathbf{1}_{[0,\infty)]}(x), \tag{D.8}$$

*where*

$$Q(-\infty) = 1 - \int_0^\infty \frac{2}{\sqrt{2\pi\sigma^2}} e^{\frac{-(t+L)^2}{2\sigma^2}} \, \mathrm{d}x.$$

*Then, we have that the PLRV $\omega$,*

$$\omega = \log \frac{P(x)}{Q(x)}, \quad x \sim P, \tag{D.9}$$

*is distributed as*

$$\omega \sim \frac{L^2}{2\sigma^2} + \left|\mathcal{N}\left(0, \frac{L^2}{\sigma^2}\right)\right|.$$

*Proof.* As $P$ has its support on $[0, \infty)$, we need to consider the values of the privacy loss function $\log \frac{P(x)}{Q(x)}$ only on $[0, \infty)$. We have, for all $x \geq 0$,

$$\log \frac{P(x)}{Q(x)} = \frac{L}{\sigma^2} \cdot x + \frac{L^2}{2\sigma^2}.$$

Since $x \sim P$, we see that $\frac{L}{\sigma^2} \cdot x \sim \left|\mathcal{N}\left(0, \frac{L^2}{\sigma^2}\right)\right|$ and the claim follows. $\square$

*Remark D.5.* In Lemma D.4, instead of shifting part of the mass of $P$ to $-\infty$ when forming $Q$, we could place this mass anywhere on the negative real axis. This would not affect the PLRV $\omega$.

We can shift the PLRV $\omega$ given by Lemma D.4 by scaling the distribution $Q$. We get the following.

*Lemma D.6. Let $\sigma > 0$ and $L > 0$. Suppose $P$ is the density function given in Eq. (D.7) and $Q$ the density function*

$$Q(x) = Q(-\infty) \cdot \delta_{-\infty}(x) + e^{-\left|\log\left(1-\frac{\beta}{\lambda}\right)\right|} \cdot \frac{2}{\sqrt{2\pi\sigma^2}} e^{\frac{-(t-L)^2}{2\sigma^2}} \mathbf{1}_{[0,\infty)]}(x), \tag{D.10}$$

*where*

$$Q(-\infty) = \left(1 - \mathrm{e}^{-\left|\log\left(1 - \frac{\beta}{\lambda}\right)\right|} \cdot \int_0^\infty \frac{2}{\sqrt{2\pi\sigma^2}} \mathrm{e}^{\frac{-(t-L)^2 \, 2}{2\sigma}} \, \mathrm{d}x\right).$$

*Then, the PLRV $\omega$ determined by $P$ and $Q$ is distributed as*

$$\omega \sim \left|\log\left(1 - \frac{\beta}{\lambda}\right)\right| + \frac{L^2}{2\sigma^2} + \left|\mathcal{N}\left(0, \frac{L^2}{\sigma^2}\right)\right|.$$

*Proof.* Showing this goes as the proof of Lemma D.4. We just now have that for all $x \geq 0$:

$$\log\frac{P(x)}{Q(x)} = \left|\log\left(1 - \frac{\beta}{\lambda}\right)\right| + \frac{L^2}{2\sigma^2} + \frac{L}{\sigma^2} \cdot x.$$

$\square$

As a corollary of Lemma D.6 and Thm. 2.4, we have:

*Lemma D.7. Let $k \in \mathbb{Z}_+$ and let for each $i \in [k]$*

$$\omega_i \sim \left|\log\left(1 - \frac{\beta}{\lambda}\right)\right| + \frac{L^2}{2\sigma^2} + \left|\mathcal{N}(0, \frac{L^2}{\sigma^2})\right|,$$

*such that $\omega_i$'s are independent. Then, the $k$-wise adaptive composition of $\hat{\theta}(Z)$ is $(\epsilon, \delta(\epsilon))$-DP for*

$$\delta(\epsilon) = \mathbf{E}_{s\sim\omega_1+\ldots+\omega_k}[1 - e^{\epsilon-s}]_+. \tag{D.11}$$

### D.3.1 Numerical Evaluation of $(\epsilon, \delta)$-Bounds for Compositions

Figure 3 shows the result of applying the FFT-based numerical method of Koskela et al. (2021) for evaluating the expression (D.11). We compare the resulting approximate DP bounds to those obtained from the RDP bounds combined with standard composition results (Mironov, 2017).

Notice that we could also carry out tighter accounting of the approximative minima perturbation (Section 4) by adding the PLRVs of the Gaussian mechanism to the total PLRV, similarly as RDP parameters of the Gaussian mechanism are added to the RDP guarantees of the objective perturbation mechanism (Theorem 4.1). Adding the Gaussian PLRV to the total PLRV using convolutions is straightforward using the method of Koskela et al. (2021).

### D.4 Auxiliary Lemma

For Theorem 3.1, we need the following auxiliary result.

*Lemma D.8. Denote $f_{\mu,\sigma^2}(x)$ the density function of the normal distribution $\mathcal{N}(\mu, \sigma^2)$ and let $c \geq \mu$. Let $g(x)$ be a non-negative differentiable non-decreasing function on $[c, \infty)$. Then, if $\sigma_1 \leq \sigma_2$,*

$$\int_c^\infty f_{\mu,\sigma_1^2}(x) \cdot g(x) \, \mathrm{d}x \leq \int_c^\infty f_{\mu,\sigma_2^2}(x) \cdot g(x) \, \mathrm{d}x.$$

*Proof.* By integration by parts, we have

$$\int_c^\infty f_{\mu,\sigma^2}(x) \cdot g(x) \, \mathrm{d}x = -\Phi_{\mu,\sigma^2}(c) \cdot g(c) - \int_c^\infty \Phi_{\mu,\sigma^2}(x) \cdot g'(x) \, \mathrm{d}x, \tag{D.12}$$

where $\Phi_{\mu,\sigma^2}(x)$ denotes the cdf of $\mathcal{N}(\mu, \sigma^2)$. A simple calculation shows that for all $x \in \mathbb{R}$,

$$\frac{\partial}{\partial\sigma}\Phi_{\mu,\sigma^2}(x) = -\frac{x-\mu}{\sigma^2}f_{\mu,\sigma^2}(x).$$

Thus, $\Phi_{\mu,\sigma^2}(c)$ is a non-increasing function of $\sigma$ for all $c \geq \mu$. Furthermore, the first term in (D.12) is a non-decreasing function of $\sigma$ since $g(c)$ is non-negative and the second term is a non-decreasing function of $\sigma$, since $g'(x)$ is non-negative for all $x \in [c, \infty)$. $\square$

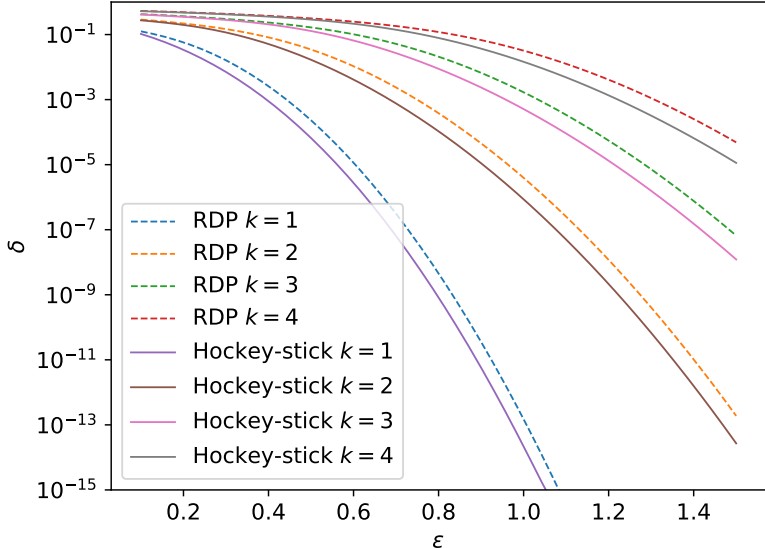

Figure 3: Comparison of our RDP bound (implied $(\epsilon, \delta)$-DP bound) and our numerical PLRV bound (D.11) for different numbers of compositions $k$, when $\sigma = 8.0$, $\beta = 1.0$ and $\lambda = 10.0$.

# E RDP guarantee of Algorithm 1

In what follows, we will present a (corrected) privacy guarantee for Approximate Minima Perturbation (i.e., Algorithm 1 without gradient clipping). We will then demonstrate that the "clipped-gradient" function $\ell_C(\theta)$ not only bounds the per-example gradient norm by $C$, but also preserves other properties (i.e., $\beta$-smoothness and GLM structure) required for the privacy guarantees stated in Theorem 3.2.

## E.1 Privacy Guarantee for Approximate Minima Perturbation

The proof of the privacy guarantee for Approximate Minima Perturbation (Iyengar et al., 2019), i.e. Algorithm 1 without gradient clipping, falls prey to the same trap as previous work on objective perturbation. In particular, we see that there is a mistake in Lemma IV.1, with the assertion that "we get the statement of the lemma from the guarantees of the Gaussian mechanism." The Gaussian mechanism is inapplicable in Lemma IV.1 for similar reasons as discussed in Section G.

The proof of Theorem E.1 corrects this issue. We state it in terms of RDP, but it can also extend to approximate DP and other DP variants.

---

**Algorithm 2** Approximate Minima Perturbation (Iyengar et al., 2019)

---

**Input:** dataset $Z$; noise levels $\sigma, \sigma_{\text{out}}$; $\beta$-smooth loss function $\ell(\cdot)$ with Lipschitz constant $L$; regularization strength $\lambda$; gradient norm threshold $\tau$.

Sample $b \sim \mathcal{N}(0, \sigma^2 I_d)$.

Let $\mathcal{L}^P(\theta; Z) = \sum_{z \in Z} \ell(\theta; z) + \frac{\lambda}{2} ||\theta||_2^2 + b^T \theta$.

Solve for $\tilde{\theta}$ such that $||\nabla \mathcal{L}_C^P(\tilde{\theta}; Z)||_2 \leq \tau$.

Output $\tilde{\theta}^P = \tilde{\theta} + \mathcal{N}(0, \sigma_{\text{out}}^2 I_d)$.

---

*Theorem* E.1 (RDP guarantees of Approximate Minima Perturbation). *Consider the Approximate Minima Perturbation algorithm which satisfies $(\alpha, \epsilon)$-RDP for any $\alpha > 1$ with*

$$\epsilon \leq -\log\left(1 - \frac{\beta}{\lambda}\right) + \frac{L^2}{2\sigma^2} + \frac{1}{\alpha - 1} \log \mathbb{E}_{X \sim \mathcal{N}\left(0, \frac{L^2}{\sigma^2}\right)} \left[e^{(\alpha - 1)|X|}\right] + \frac{\left(\frac{2\tau}{\lambda}\right)^2 \alpha}{2\sigma_{out}^2}.$$

*Proof.* Sample $b \sim \mathcal{N}(0, \sigma^2 I_d)$ and let $\mathcal{L}^P(\theta; Z, b) := \sum_{z \in Z} \ell(\theta; z) + \frac{\lambda}{2} \|\theta\|_2^2 + b^T \theta$, i.e., the perturbed and regularized objective function used by ObjPert.

Let $\theta^P = \arg\min \mathcal{L}^P(\theta; Z, b)$. From Chaudhuri et al. (2011); Kifer et al. (2012) we know that there is a bijection $b(\theta^P; Z)$ from the output $\theta^P$ to the noise vector $b$.

Consider a blackbox algorithm $\theta^A(Z, b)$ which returns $\theta$ such that $\|\nabla \mathcal{L}^P(\theta; Z, b)\| \leq \tau$. Define query

$$q(Z, \theta^P) = \theta^A(Z) - \theta^P.$$

where $\theta^A(Z)$ is an abbreviation for $\theta^A(Z, b(\theta^P; Z))$.

We assume that $q$ can recover $b$ from the input $\theta^P$ via the bijection $b(\theta^P; Z)$, and hence has access to the perturbed objective function $\mathcal{L}^P(\theta; Z, b)$.

Notice that since $\mathcal{L}^P$ is $\lambda$-strongly convex, by applying the Cauchy-Schwarz inequality and by Definition K.3 we see that for any $\theta_1, \theta_2$,

$$\left\| \nabla \mathcal{L}^P(\theta_1) - \nabla \mathcal{L}^P(\theta_2) \right\|_2 \|\theta_1 - \theta_2\|_2 \geq \left( \nabla \mathcal{L}^P(\theta_1) - \nabla \mathcal{L}^P(\theta_2) \right)^T (\theta_1 - \theta_2) \geq \lambda \|\theta_1 - \theta_2\|_2^2.$$

Algorithm $\theta^A(Z, b)$ guarantees that its output $\theta$ satisfies $\|\nabla \mathcal{L}^P(\theta)\|_2 \leq \tau$ and by first-order conditions on the perturbed objective function, $\nabla \mathcal{L}^P(\theta^P; Z, b) = 0$. It follows that for any dataset $Z$ and $\theta^P$,

$$\left\| \theta^A(Z) - \theta^P \right\|_2 \leq \frac{\tau}{\lambda},$$

Since the algorithm $\theta^A(Z, b)$ guarantees that $\|\theta^A - \theta^P\|_2 \leq \gamma/\lambda$, then conditioning on $\theta^P$, $q(Z, \theta^P)$ has a global sensitivity bounded by $2\gamma/\lambda$ since

$$\|q(Z, \theta^P) - q(Z', \theta^P)\| \leq \|(\theta^A(Z) - \theta^P) - (\theta^A(Z') - \theta^P)\|_2 \leq \|(\theta^A(Z) - \theta^P)\| + \|(\theta^A(Z') - \theta^P)\| \leq \frac{2\gamma}{\lambda}.$$

Now, the algorithm that first draws $b$ then outputs $\theta^A(Z, b) + \mathcal{N}(0, \sigma^2 I_d)$ is equivalent to

- First run ObjPert that returns $\theta^P$.

- Release $\hat{\Delta} = q(Z, \theta^P) + \mathcal{N}(0, \sigma^2 I_d)$.

- Return $\theta^P + \hat{\Delta}$.

This is adaptive composition of ObjPert with the Gaussian mechanism. The third step is post processing.

The privacy guarantee stated in Theorem E.1 is thus achieved by combining the results of Theorem 3.2 (RDP of ObjPert), Theorem B.1 (RDP of the Gaussian mechanism) with $\Delta_q = \frac{2\tau}{\lambda}$, and Lemma K.5 (adaptive composition for RDP mechanisms).

$\square$

## E.2 The "Clipped-Gradient" Function

The RDP guarantees of objective perturbation (stated in Theorem 3.2) require several assumptions on the loss function $\ell(\theta; Z)$. If we can demonstrate that these properties are satisfied by the "clipped-gradient" loss function $\ell_C(\theta; Z)$, then the rest of the proof of Theorem 4.1 (the privacy guarantee of Algorithm 1) will follow directly from that.

In particular, we need to show:

1. That $\ell_C(\theta; z)$ retains the convex GLM structure of the original function $\ell(\theta; z)$.

2. That $\ell_C(\theta; z)$ satisfies $\|\nabla \ell_C(\theta; z)\|_2 \leq C$ for any $\theta, z$.

3. That $\ell_C(\theta; z)$ has the same $\beta$-smoothness parameter as the original function $\ell(\theta; z)$.

4. That even though $\ell_C(\theta; z)$ is not twice-differentiable everywhere, the privacy guarantees of objective perturbation (whose proof involves a Jacobian mapping) still hold.

We will begin by stating a result from Song et al. (2020).

**Theorem E.2** (Song et al., 2020, Lemma 5.1). *Let $f : \mathbb{R} \to \mathbb{R}$ be any convex function and let $C \in \mathbb{R}_+$ be any positive value. For any non-zero $x \in \mathbb{R}^d$, define*

$$U_L = \left\{ u : g < -\frac{C}{\|x\|_2} \quad \forall g \in \partial f(u) \right\},$$

$$U_H = \left\{ u : g > \frac{C}{\|x\|_2} \quad \forall g \in \partial f(u) \right\}.$$

*If $U_L$ is non-empty, let $u_L = \sup U_L$; otherwise $u_L = -\infty$. If $U_H$ is non-empty, let $u_H = \inf U_H$; otherwise $u_H = \infty$. For any non-zero $x \in \mathbb{R}^d$, let*

$$f_C(u) = \begin{cases} -\frac{C}{\|x\|_2} (u - u_L), & \text{for } u \in (-\infty, u_L) \\ f(u; y), & \text{for } u \in (u_L, u_H) \\ \frac{C}{\|x\|_2} (u - u_H), & \text{for } u \in (u_H, \infty) \end{cases}$$

*Define $u_x(\theta) = x^T \theta$. Then the following holds.*

1. *$f_C$ is convex.*

2. *Let $\ell(\theta; (x, y)) = f(u_x(\theta); y)$ for any $\theta, z = (x, y)$. Then we have*

$$\partial_\theta \ell_C(\theta; z) = \left\{ \min \left\{ 1, \frac{C}{\|u_x(\theta)\|_2} \right\} \cdot u : u \in \partial_\theta \ell(\theta; z) \right\}.$$

The first two desired properties of $\ell_C(\theta; z)$, i.e. GLM structure and gradient norm bound $C$, follow directly from the above theorem. Next, we will prove the third property of $\beta$-smoothness.

**Theorem E.3.** *For a data point $z = (x, y)$, consider a function $f$ such that $\ell(\theta; z) = f(x^T \theta; y)$. Suppose that $f(x^T \theta; y)$ satisfies $\beta$-smoothness. Then the "clipped-gradient" function $f_C(x^T \theta; y)$ defined in Lemma 5.1 of Song et al. (2020) also satisfies $\beta$-smoothness.*

*Proof.* Because $f$ is $\beta$-smooth by assumption, we know that $f(u)$ satisfies $\beta$-smoothness for all $u \in (u_L, u_H)$. When $u \in (-\infty, u_L)$ or when $u \in (u_H, \infty)$, the function $f_C(u)$ is linear in $u$ and thus is 0-smooth (hence satisfying $\beta$-smoothness). $\square$

Lastly, the proof of objective perturbation (see, e.g., Theorem 9 of Chaudhuri et al. (2011)) requires that the loss function be twice-differentiable. Even though $\ell_C(\theta; z)$ is not twice-differentiable everywhere, the privacy guarantees of objective perturbation still hold. To show this, we can invoke Corollary 13 of Chaudhuri et al. (2011). This corollary assumes the Huber loss; for brevity, we will leave it as an exercise for the reader to verify that the proof also carries through for the "clipped-gradient" loss.

# F    Computational Guarantee of Algorithm 1

In this section, we provide a *computational guarantee* to Algorithm 1 in terms of the number of gradient evaluations on individual loss functions to compute the approximate minimizer for achieving (up to a constant) the information-theoretical limit.

Let $f(\theta) := \sum_i \ell_i(\theta) + \frac{\lambda}{2} \|\theta\|^2 + b^T \theta$, i.e., the perturbed and regularized objective function used by ObjPert. Let $\theta^{**}$ be the output returned by the blackbox algorithm $\theta^A(\cdot)$ described in Section E. Note the deviation from the notation used in the previous section.

Iyengar et al. (2019) proposed a procedure that keeps checking the gradients in an iterative optimization algorithm and stops when the gradient is smaller than $\tau$. This always ensures that $\|\nabla f(\theta^{**})\| \leq \tau$.

And using tools from the next section, it can be proven that it implies that $\|\theta^{**} - \theta^*\|_2 \leq \tau/\lambda$ as was previously stated. But how many iterations it takes for this to happen for specific algorithms was not explicitly considered.

## F.1 Tools from convex optimization

We will need a few tools from convex optimization.

Firstly, under our assumption that $\ell_i$ is $\beta$-smooth, $f$ is $n\beta + \lambda$-smooth and $\lambda$-strongly convex. Let $L := n\beta + \lambda$ as a shorthand.

By $L$-smoothness (gradient Lipschitzness), and the optimality of $\theta^*$, we have that for any $\theta$

$$\|\nabla f(\theta)\| = \|\nabla f(\theta) - \nabla f(\theta^*)\| \leq L\|\theta - \theta^*\|_2. \tag{F.1}$$

By $\lambda$-strong convexity, we get

$$f(\theta) - f^* \geq \frac{\lambda}{2}\|\theta - \theta^*\|^2 \geq \frac{\lambda}{2L^2}\|\nabla f(\theta)\|^2 \tag{F.2}$$

By strong convexity also implies that

$$\|\nabla f(\theta)\| \geq \lambda\|\theta - \theta^*\| \tag{F.3}$$

which is the quantity used to establish the global sensitivity of $q(D, \theta^*)$ as we talked about earlier.

(F.2) and (F.3) sandwich $\|\theta - \theta^*\|$ in between by

$$\frac{\|\nabla f(\theta)\|}{L} \leq \|\theta - \theta^*\| \leq \frac{\|\nabla f(\theta)\|}{\lambda}.$$

## F.2 Computational bounds for Stopping at small gradient

(F.1) and (F.2) together provides bounds for $\|\nabla f(\theta)\|$ using either objective function or argument convergence (in square $\ell_2$.)

$$\|\nabla f(\theta)\|^2 \leq \min\left\{\frac{2L^2}{\lambda}(f(\theta) - f^*), L^2\|\theta - \theta^*\|^2\right\}.$$

Standard convergence results are often parameterized in terms of either suboptimality $f(\theta) - f^*$ or argument $\|\theta - \theta^*\|^2$. In the following we instantiate specific convergence bounds for deriving computation guarantees.

**Gradient Descent.** If we run gradient descent with learning rate $1/L$ for $T$ iterations from $\theta_0$, then

$$\|\theta_T - \theta^*\|^2 \leq (1 - \frac{\lambda}{L})^T\|\theta_0 - \theta^*\|^2$$

which implies that

$$\|\nabla f(\theta_T)\|^2 \leq L^2(1 - \frac{\lambda}{L})^T\|\theta_0 - \theta^*\|^2$$

This happens deterministically (with no randomness, or failure probability).

One may ask why are we not running a fixed number of iterations and directly applying the bound to $\|\theta_T - \theta^*\|$ in order to control the $\ell_2$ sensitivity. That works fine, except that we have an unconstrained problem and $\theta^*$ can be anywhere, thus there might not be a fixed parameter $T$ to provide a required bound for all input $\theta^*$. We also do not know where $\theta^*$ is during the actual execution of the algorithm and thus cannot compute $\|\theta_0 - \theta^*\|$ directly.

The "gradient-norm check" as a stopping condition from Iyengar et al. (2019) is nice because it always ensures DP for any $\theta^*$ (at a price of sometimes running for a bit longer).

To ensure $\|\nabla f(\theta)\| \leq \gamma$, the number of iterations

$$T = \frac{\log(\frac{L^2\|\theta_0 - \theta^*\|^2}{\gamma^2})}{\log(1 + \frac{\lambda}{L-\lambda})} \leq \frac{2(L-\lambda)}{\lambda}\log(\frac{L^2\|\theta_0 - \theta^*\|^2}{\gamma^2}) = \frac{2n\beta}{\lambda}\log(\frac{(n\beta + \lambda)^2\|\theta_0 - \theta^*\|^2}{\gamma^2})$$

Since each gradient computation requires $n$ incremental gradient evaluation, under the regime that $\lambda$ is independent of $n$, under the choice that $\lambda = 1/\epsilon$ independent to $n$ from the standard calibration, the total number of is therefore $O(n^2 \log n)$ for achieving $\gamma \leq n^{-v}$ for any constant $v > 0$.

The quadratic runtime is not ideal, but it can be improved using accelerated gradient descent which gives a convergence bound of

$$f(\theta_T) - f^* \leq (1 - \sqrt{\frac{\lambda}{L}})^T \|\theta_0 - \theta^*\|^2.$$

This would imply a computational guarantee of $O(n^{1.5} \log n)$. The overall computation bound depends on $\|\theta^*\|$ which is random (due to objective perturbation). The dependence on $\|\theta^*\|$ is only logarithmic though.

**Finite Sum and SAG.** The result can be further improved if we uses stochastic gradient methods. However, the sublinear convergence of the standard SGD or its averaged version makes the application of the above conversion rules somewhat challenging.

By taking advantage of the finite sum structure of $f(\theta)$ one can obtain faster convergence.

First of all, the finite sum structure says that $f(\theta) = \sum_{i=1}^{n} f_i(\theta)$. In our case, we can split the regularization and linear perturbation to the $n$ data points, i.e.,

$$f_i(\theta) = \ell_i(\theta) + \frac{\lambda}{2n} \|\theta\|^2 + \frac{b^T \theta}{n}.$$

Check that it satisfies $\beta + \lambda/n$ smoothness.

There is a long list of methods that satisfy the faster convergence for finite sum problems, e.g., SAG, SVRG, SAGA, SARAH and so on (see, e.g., Nguyen et al., 2022, for a recent survey). Specifically, Stochastic Averaged Gradient (Schmidt et al., 2017) (and similarly others with slightly different parameters) satisfies

$$\mathbb{E}\left[f(\theta^T) - f^*\right] \leq (1 - \min\{\frac{\lambda}{16L}, \frac{1}{8n}\})^T \cdot (\frac{3n}{2}(f(\theta_0) - f^*) + 4L\|\theta_0 - \theta^*\|^2).$$

Therefore, by (F.2), we have

$$\mathbb{E}\left[\|\nabla f(\theta_T)\|^2\right] \leq (1 - \min\{\frac{\lambda}{16L}, \frac{1}{8n}\})^T \cdot \frac{L^2}{\lambda}(\frac{3n}{2}(f(\theta_0) - f^*) + 4L\|\theta_0 - \theta^*\|^2).$$

Note that each iteration costs just one incremental gradient evaluation, so to ensure $\mathbb{E}[\|\nabla f(\theta_T)\|^2] \leq \gamma^2$, the computational complexity is on the order of

$$\max\{n, \frac{L}{\lambda}\} \log \left(\frac{nL \max\{f(\theta_0) - f^*, \|\theta_0 - \theta^*\|^2\}}{\lambda \gamma}\right)$$

This is $O(n \log n)$ runtime to any $\gamma = n^{-s}$ for a constant $s > 0$.

On the other hand, the main difference from the gradient descent result is that we only get convergence in expectation. By Markov's inequality

$$\mathbb{P}\left[\|\nabla f(\theta_T)\|^2 > \gamma^2\right] \leq \frac{(1 - \min\{\frac{\lambda}{16L}, \frac{1}{8n}\})^T \cdot \frac{L^2}{\lambda}(\frac{3n}{2}(f(\theta_0) - f^*) + 4L\|\theta_0 - \theta^*\|^2)}{\gamma^2} := \delta,$$

which implies high probability convergence naturally.

*Theorem* F.1. *Assume $\lambda \geq \beta$. The algorithm that runs SAG and checks the stopping condition $\|\nabla f(\theta_T)\| \leq \gamma$ after every $n$ iteration will terminate with probability at least $1 - \delta$ in less than*

$$C \max\{n, \frac{n\beta}{\lambda}\} \log \left(\frac{n\beta \max\{\|\theta_0 - \theta^*\|, (f(\theta_0) - f^*)\}}{\gamma\delta}\right)$$

*incremental gradient evaluations, where $C$ is a universal constant.*

**How to set $\gamma$ to achieve information-theoretic limit?** The lower bounds for convex and smooth losses in differentially private ERM are well-known (Bassily et al., 2014) and it is known that among GLMs, $\theta^*$ from ObjPert achieves the lower bound with appropriate choices of $\lambda, \sigma$. Notably, $\lambda \asymp d/\epsilon$ for achieving an $(\epsilon, \delta)$-DP.

$$\mathbb{E}[\sum_i \ell_i(\theta^*)] - \min_\theta \sum_i \ell_i(\theta) \leq \text{MinimaxExcessEmpirialRisk}$$

Let $\hat{\theta} = \theta_T + N(0, \frac{\gamma^2}{2\lambda^2\rho}I)$ be the final output.

By the $nG$ Lipschitzness of $\sum_i \ell_i$, with high probability over the Gaussian mechanism, we have that

$$\mathbb{E}[\sum_i \ell_i(\hat{\theta})] - \sum_i \ell_i(\theta^*) \le nG(\|\theta_T - \theta^*\| + \|\hat{\theta} - \theta_T\|) \le nG\gamma(1 + \frac{\sqrt{\frac{d\log d}{\rho}}}{\lambda})$$

where $\rho$ is the zCDP parameter for the Gausssian mechanism chosen to match the large $\alpha$ part of the ObjectivePerturbation's RDP bound, which increases the overall RDP by $\alpha\rho$.

Thus, it suffices to take $\gamma = \text{MinimaxExcessEmpirialRisk}/(nG(1 + \frac{\sqrt{d\log d/\rho}}{\lambda}))$.

To conclude, the above results imply that the computationally efficient objective perturbation achieves the optimal rate under the same RDP guarantee with an algorithm that terminates in $O(n\log n)$ time with high probability.

# G  The GLM Bug

## G.1  Discussion

Limiting our main results to generalized linear models might appear restrictive — but we argue that the GLM assumption is not specific to our paper, but rather has been lurking in the objective perturbation literature for some time now.

Let's first take a look at Section 3.3.2 of Chaudhuri et al. (2011): Lemma 10 requires that the matrix $E$ have rank at most 2, but this is not necessarily true without assuming GLM structure. This is used to bound the determinant of the Jacobian, and corresponds to the first term of our bound in Theorem 3.2.

It is a similar story for bounding the log ratio / difference between the noise vector densities under neighboring datasets, corresponding to the second and third terms of our bound in Theorem 3.2. Let's also revisit this line from the proof of Lemma 17 of the Kifer et al. (2012) paper: "Note that $\Gamma$ is independent of the noise vector." This is not true without assuming GLM structure! (In their proof, $\Gamma$ is the difference between the noise vectors under neighboring distributions. From first-order conditions at the minimizer of the perturbed objective, we can see that $\Gamma = \nabla \ell(\theta^P)$, where $\theta^P$ is a function of the noise vector $b$.

In fact, to our knowledge, Iyengar et al. (2019) was the first work to acknowledge the GLM assumption on objective perturbation. But their privacy proof also fails to handle the dependence on the noise vector! In Theorems 3.2 and 3.1, we have included a careful analysis including a discussion on how the GLM assumption removes this dependence.

## G.2  RDP bound for non-GLMs

In this section we generalize the RDP bound for objective perturbation to a general class of smooth convex losses.

We first state the following bound from Theorem 6

*Theorem G.1.*

$$\epsilon_1(\hat{\theta}^P, D, D_{\pm z}) = \left| -\log \prod_{j=1}^d \left(1 \mp \mu_j\right) + \frac{1}{2\sigma^2}\|\nabla\ell(\hat{\theta}^P; z)\|_2^2 \pm \frac{1}{\sigma^2}\nabla J(\hat{\theta}^P; D)^T \nabla\ell(\hat{\theta}^P; z) \right|,$$

*where* $\mu_j = \lambda_j u_j^T \left(\nabla b(\hat{\theta}^P; D) \mp \sum_{k=1}^{j-1} \lambda_k u_k u_k^T\right)^{-1} u_j$ *according to the eigendecomposition* $\nabla^2\ell(\theta; z) = \sum_{k=1}^d \lambda_k u_k u_k^T$.

*Theorem G.2* (RDP bound for non-GLMS.). *Let* $\|\nabla\ell(\theta; z)\|_2 \le L$ *and* $\nabla^2\ell(\theta; z) \prec \beta I_d$ *for all* $\theta \in \Theta$ *and* $z \in \mathcal{X} \times \mathcal{Y}$. *Objective perturbation satisfies* $(\alpha, \epsilon)$-*RDP for any* $\alpha > 1$ *with*

$$\epsilon = -d\log\left(1 - \frac{\beta}{\lambda}\right) + \frac{L}{2\sigma^2} + \frac{1}{\alpha-1}\log \mathbb{E}_{Z\sim\chi_d}\left[e^{(\alpha-1)\frac{L^2}{\sigma^2}Z}\right],$$

*where $Z \sim \chi_d$ if $Z = \sqrt{\sum_{i=1}^d X_i^2}$ and $X_i \sim \mathcal{N}(0,1)$ for all $i \in [d]$.*

*Proof.* The above follows from Theorem 6 in (Redberg & Wang, 2021) applied to the analysis of Theorem 3.2. We find that

$$\left| -\log \prod_{j=1}^d (1 \mp \mu_j) \right| = \left| -\sum_{j=1}^d \log(1 \mp u_j) \right| \le -\sum_{j=1}^d \log(1 - u_j) \le -d\log\left(1 - \frac{\beta}{\lambda}\right).$$

We now need to bound

$$\frac{1}{\alpha - 1} \log \mathbb{E}\left[ e^{(\alpha-1)\frac{1}{\sigma^2}\nabla J(\theta^P)^T \nabla \ell(\theta^P)} \right].$$

Recall that $b \sim \mathcal{N}(0, \sigma^2 I_d)$. Using the first-order condition $\nabla J(\theta^P) = -b$ and the Cauchy-Schwarz inequality, we have

$$-b^T \nabla \ell(\theta^P) \le \left| -b^T \nabla \ell(\theta^P) \right| \le L||b||_2.$$

So

$$\frac{1}{\alpha - 1} \log \mathbb{E}\left[ e^{(\alpha-1)\frac{1}{\sigma^2}\nabla J(\theta^P)^T \nabla \ell(\theta^P)} \right] \le \frac{1}{\alpha - 1} \log \mathbb{E}_{Z \sim \chi_d}\left[ e^{(\alpha-1)\frac{L}{\sigma^2}Z} \right].$$

$\square$

# H    Excess Empirical Risk of Algorithm 1

Our goal in this section is to find a bound on the excess empirical risk:

$$\mathbb{E}\left[ \mathcal{L}(\tilde{\theta}^P; Z) \right] - \mathcal{L}(\theta^*; Z).$$

*Theorem H.1. Let $\tilde{\theta}^P$ be the output of Algorithm 1 and $\theta^* = \arg\min \mathcal{L}(\theta)$ the minimizer of the loss function $\mathcal{L}(\theta) = \sum_{i=1}^n \ell(\theta; z_i)$. Denote $||\mathcal{X}||$ as the diameter of the set $\mathcal{X}$. We have*

$$\mathbb{E}\left[ \mathcal{L}(\tilde{\theta}^P; Z) \right] - \mathcal{L}(\theta^*; Z) \le nL\left(\frac{\tau}{\lambda} + \sigma_{out}\sqrt{d}\right) + \frac{(n\beta||\mathcal{X}||_2^2 + \lambda)d\sigma^2}{2\lambda^2} + \frac{\lambda}{2}||\theta^*||_2^2.$$

*Proof.* Following the proof of Theorem 2 from Iyengar et al. (2019) (itself adapted from Kifer et al. (2012)), we write

$$\mathcal{L}(\tilde{\theta}^P) - \mathcal{L}(\theta^*) = \left(\mathcal{L}(\tilde{\theta}^P) - \mathcal{L}(\theta^P)\right) + \left(\mathcal{L}(\theta^P) - \mathcal{L}(\theta^*)\right).$$

By the $\lambda$-strong convexity of $\mathcal{L}^P$, for any $\tilde{\theta}, \theta^*$ we have

$$\left(\nabla\mathcal{L}^P(\tilde{\theta}) - \nabla\mathcal{L}^P(\theta^P)\right)^T \left(\tilde{\theta} - \theta^P\right) \ge \lambda||\tilde{\theta} - \theta^P||_2^2.$$

By first-order conditions, $\nabla\mathcal{L}^P(\theta^P) = 0$. Applying the Cauchy-Schwarz inequality along with our stopping criteria on the gradient norm, we then have

$$||\tilde{\theta} - \theta^P||_2 \le \frac{1}{\lambda}||\nabla\mathcal{L}^P(\tilde{\theta})||_2 \le \frac{\tau}{\lambda}.$$

Let $b_2 \sim \mathcal{N}(0, \sigma_{out}^2 I_d)$. Because $\mathcal{L}$ is $nL$-Lipschitz continuous, we have

$$\begin{aligned}
\left(\mathcal{L}(\tilde{\theta}^P) - \mathcal{L}(\theta^P)\right) &\le nL||\tilde{\theta}^P - \theta^P||_2 \\
&= nL||\tilde{\theta} + b_2 - \theta^P||_2 \\
&\le nL\left(||\tilde{\theta} - \theta^P||_2 + ||b_2||_2\right) \\
&\le nL\left(\frac{\tau}{\lambda} + ||b_2||_2\right).
\end{aligned}$$

By Lemma K.10,

$$\mathbb{E}\left[nL\left(\frac{\tau}{\lambda}+||b_2||_2\right)\right] \leq nL\left(\frac{\tau}{\lambda}+\sigma_{out}\sqrt{d}\right).$$

To bound the expectation of $\mathcal{L}(\theta^P) - \mathcal{L}(\theta^*)$, we can write

$$\mathcal{L}(\theta^P) - \mathcal{L}(\theta^*) = \left(\mathcal{L}(\theta^P) - \mathcal{L}_\lambda(\theta^P)\right) + \left(\mathcal{L}_\lambda(\theta^P) - \mathcal{L}_\lambda(\theta^*_\lambda)\right) + \left(\mathcal{L}_\lambda(\theta^*_\lambda) - \mathcal{L}_\lambda(\theta^*)\right) + \left(\mathcal{L}_\lambda(\theta^*) - \mathcal{L}(\theta^*)\right).$$

Observe that

$$\mathcal{L}(\theta^P) - \mathcal{L}_\lambda(\theta^P) = -\frac{\lambda}{2}||\theta^P||_2^2 \leq 0,$$

and that by optimality $\left(\theta^*_\lambda = \arg\min_{\theta \in \mathbb{R}^d} \mathcal{L}_\lambda(\theta)\right)$, we have

$$\mathcal{L}_\lambda(\theta^*_\lambda) - \mathcal{L}_\lambda(\theta^*) \leq 0.$$

Observe also that

$$\mathcal{L}_\lambda(\theta^*) - \mathcal{L}(\theta^*) = \frac{\lambda}{2}||\theta^*||_2^2.$$

So

$$\mathcal{L}(\theta^P) - \mathcal{L}(\theta^*) \leq \mathcal{L}_\lambda(\theta^P) - \mathcal{L}_\lambda(\theta^*_\lambda) + \frac{\lambda}{2}||\theta^*||_2^2.$$

By Taylor's Theorem, for some $\theta' \in \left[\theta^P, \theta^*_\lambda\right]$ we can write

$$\mathcal{L}_\lambda(\theta^P) - \mathcal{L}_\lambda(\theta^*_\lambda) = \nabla\mathcal{L}_\lambda(\theta^*_\lambda)\left(\theta^P - \theta^*_\lambda\right)^T + \frac{1}{2}||\theta^P - \theta^*_\lambda||_{\nabla^2\mathcal{L}_\lambda(\theta')}^2$$

$$= \frac{1}{2}||\theta^P - \theta^*_\lambda||_{\nabla^2\mathcal{L}_\lambda(\theta')}^2. \tag{H.1}$$

The last equality is due to optimality conditions. We can then bound

$$\frac{1}{2}||\theta^P - \theta^*_\lambda||_{\nabla^2\mathcal{L}_\lambda(\theta')}^2 \leq \frac{1}{2}||\nabla^2\mathcal{L}_\lambda(\tilde{\theta})||_{\text{op}}||\theta^P - \theta^*_\lambda||_2^2$$

$$\leq \frac{1}{2}\left(n\beta||\mathcal{X}||_2^2 + \lambda\right)\frac{||b||_2^2}{\lambda^2},$$

where $b \sim \mathcal{N}(0, \sigma^2 I_d)$. Then taking the expectation,

$$\mathbb{E}\left[\mathcal{L}_\lambda(\theta^P)\right] - \mathbb{E}\left[\mathcal{L}_\lambda(\theta^*_\lambda)\right] \leq \frac{(n\beta||\mathcal{X}||_2^2 + \lambda)d\sigma^2}{2\lambda^2}.$$

Altogether we then have

$$\mathbb{E}\left[\mathcal{L}(\tilde{\theta}^P; Z)\right] - \mathcal{L}(\theta^*; Z) \leq nL\left(\frac{\tau}{\lambda}+\sigma_{out}\sqrt{d}\right) + \frac{(n\beta||\mathcal{X}||_2^2 + \lambda)d\sigma^2}{2\lambda^2} + \frac{\lambda}{2}||\theta^*||_2^2.$$

The optimal choice of $\lambda$ would then be $\lambda \asymp \frac{dL}{||\theta^*||_2\epsilon}$. The optimal choice of $\tau$ is discussed in Section F. $\qquad\square$

## H.1 Generalized Linear Model

With some additional assumptions and restrictions, we can get a tighter bound on $\mathbb{E}\left[\mathcal{L}_\lambda(\theta^P)\right] - \mathcal{L}_\lambda(\theta^*_\lambda)$.

We will assume GLM structure on $\ell(\cdot)$, i.e. $\ell(\theta; z) = f(x^T\theta; y)$. We will further assume boundedness: $c \leq f(x^T\theta; y) \leq C$ for some universal constants $c, C \in \mathbb{R}$. Applying Taylor's Theorem (in an argument similar to Equation C.8), we can show that for some $\theta'' \in \left[\theta^P, \theta^*_\lambda\right]$

$$\theta^P - \theta^*_\lambda = \nabla^2\mathcal{L}_\lambda(\theta'')^{-1}b.$$

Note that by the GLM assumption, the eigendecomposition of $\nabla^2 \mathcal{L}^*_\lambda(\theta)$ can be written as $X^T \Lambda(\theta) X$. Then plugging in from Equation H.1 and using the boundedness assumption on the loss $f$,

$$
\begin{aligned}
\mathcal{L}_\lambda(\theta^P) - \mathcal{L}_\lambda(\theta^*_\lambda) &= \frac{1}{2} \|\theta^P - \theta^*_\lambda\|^2_{\nabla^2 \mathcal{L}_\lambda(\theta')} \\
&= b^T \left(\nabla^2 \mathcal{L}_\lambda(\theta'')\right)^{-1} \nabla^2 \mathcal{L}_\lambda(\theta') \left(\nabla^2 \mathcal{L}_\lambda(\theta'')\right)^{-1} b \\
&= b^T \left(X^T \Lambda(\theta'')X + \lambda I_d\right)^{-1} \left(X^T \Lambda(\theta'')X + \lambda I_d\right) \left(X^T \Lambda(\theta')X + \lambda I_d\right)^{-1} b \\
&\leq b^T c^{-1} \left(X^T X + \lambda I_d\right)^{-1} C \left(X^T X + \lambda I_d\right) c^{-1} \left(X^T X + \lambda I_d\right)^{-1} b \\
&\leq \frac{C}{c^2} \|b\|^2_{(X^T X + \lambda I_d)^{-1}}
\end{aligned}
$$

Then in expectation,

$$
\mathbb{E}\left[\mathcal{L}_\lambda(\theta^P) - \mathcal{L}_\lambda(\theta^*_\lambda)\right] \leq \frac{C\sigma^2}{c^2} \mathrm{tr}\left(\left(X^T X + \lambda I_d\right)^{-1}\right).
$$

# I  Distance to Optimality

Consider the mechanism $\mathcal{M}(Z) = f(Z) + \mathcal{N}(0, \sigma^2 I_d)$, for a function $f : \mathcal{Z} \to \mathbb{R}^d$ with sensitivity $\Delta_f = L$. From Balle & Wang (2018), we know that for any neighboring datasets $Z$ and $Z'$, the privacy loss random variable of this mechanism is distributed as $\mathcal{N}\left(\frac{\Delta^2_{Z,Z'}}{2\sigma^2}, \frac{\Delta^2_{Z,Z'}}{\sigma^2}\right)$. Maximizing the Rényi divergence $D_\alpha\left(\mathcal{M}(Z) \,\|\, \mathcal{M}(Z')\right)$ over all neighboring datasets $Z \simeq Z'$ shows that the RDP for the Gaussian mechanism can be written as

$$
\begin{aligned}
\epsilon(\alpha) &= \frac{1}{\alpha - 1} \log \mathbb{E}\left[e^{(\alpha-1)\mathcal{N}\left(\frac{L^2}{2\sigma^2}, \frac{L^2}{\sigma^2}\right)}\right] \\
&= \frac{L^2}{2\sigma^2} + \frac{1}{\alpha - 1} \log \mathbb{E}\left[e^{(\alpha-1)\mathcal{N}\left(0, \frac{L^2}{\sigma^2}\right)}\right].
\end{aligned}
$$

Thus the main deviations between the RDP bound for objective perturbation and that of the Gaussian mechanism are 1) the leading term (a function of $\beta$ and $\lambda$ that vanishes as we increase the regularization) and 2) the moment-generating function of the *half-normal* (instead of normal) distribution.

Figure I plots the Rényi divergence $\epsilon(\alpha) := D_\alpha(\mathcal{M}(Z) \,\|\, \mathcal{M}(Z'))$ for the Gaussian mechanism ("normal"), the objective perturbation mechanism ("ObjPert"), and the mechanism [4] obtained by adding noise from the half-normal distribution ("half-normal").

We consider several different regimes of interest by varying the noise scale $\sigma$ and the regularization strength $\lambda$. There are several takeaways to observe:

1. The difference between the RDP for the half-normal mechanism and the RDP for the objective perturbation mechanism is due entirely to the leading term of the bound given in Theorem 3.2, which vanishes as $\lambda$ increases (Figures 4b and 4c). For smaller $\lambda$ there is a constant "start-up" gap between the half-normal and Objpert RDP curves (best displayed in Figure 4b) which disappears for larger $\alpha$, where the moments of the half-normal distribution overwhelm the contribution of the leading term of the objective perturbation RDP.

2. As $\sigma$ increases (e.g. between Figures 4c and 4a, and between Figures 4d and 4b), the half-normal curve – and therefore also the ObjPert curve – doesn't converge with the normal curve until larger $\alpha$.

# J  Bridging the Gap between Objective Perturbation and DP-SGD

## J.1  RDP of Objective Perturbation vs DP-SGD

DP-SGD (for example, $n^2$ rounds of sampled Gaussian mechanism with Poisson sampling probability $1/n$) is known to experience a phase transition in its RDP curve: for smaller $\alpha$, amplification by

---

[4]More formally, we say that the "half-normal" mechanism is $\mathcal{M}(Z) = f(Z) + |\mathcal{N}(0, \sigma^2)|$, where $f : \mathcal{Z} \to \mathbb{R}$ is the function we wish to release.

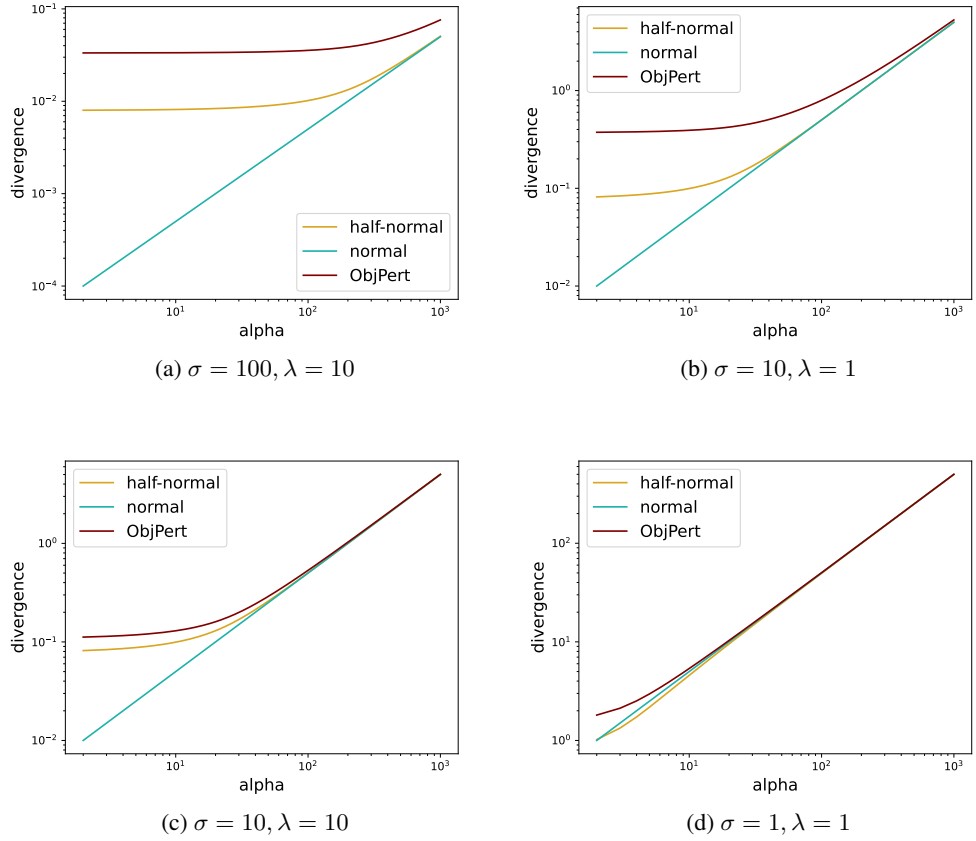

(a) $\sigma = 100, \lambda = 10$

(b) $\sigma = 10, \lambda = 1$

(c) $\sigma = 10, \lambda = 10$

(d) $\sigma = 1, \lambda = 1$

Figure 4

sampling is effective, and the RDP of a DP-SGD mechanism behaves like a Gaussian mechanism with $\epsilon(\alpha) = O(\frac{\alpha}{2\sigma^2})$, then it leaps up at a certain $\alpha$ and begins converging to $\epsilon(\alpha) = O(\frac{n^2\alpha}{2\sigma^2})$ that does not benefit from sampling at all (Wang et al., 2019; Bun et al., 2018). In contrast, the RDP curve for objective perturbation defined by Theorem 3.2 converges to the RDP curve of the Gaussian mechanism $\epsilon(\alpha) = O(\frac{\alpha}{2\sigma^2})$ *after* a certain point. Whereas DP-SGD offers stronger privacy parameters for small $\alpha$, objective perturbation is stronger for large $\alpha$, which offers stronger privacy protection for lower-probability events (see, e.g., Mironov, 2017, Proposition 10).

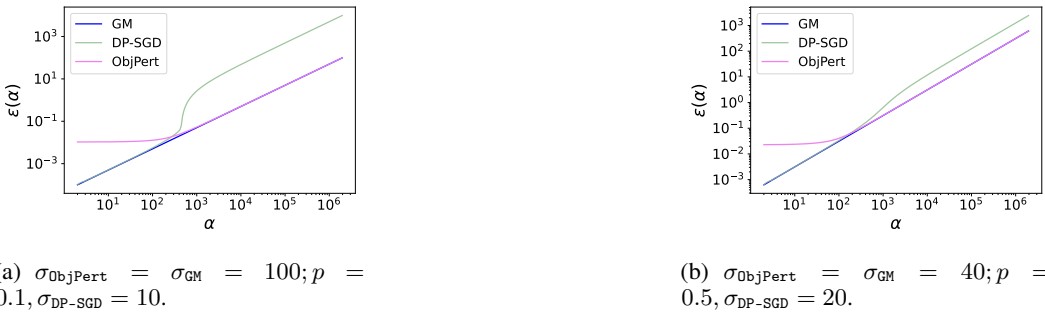

(a) $\sigma_{\texttt{ObjPert}} = \sigma_{\texttt{GM}} = 100; p = 0.1, \sigma_{\texttt{DP-SGD}} = 10$.

(b) $\sigma_{\texttt{ObjPert}} = \sigma_{\texttt{GM}} = 40; p = 0.5, \sigma_{\texttt{DP-SGD}} = 20$.

Figure 5: RDP curves of objective perturbation and DP-SGD.

## J.2 A Spectrum of DP Learning Algorithms

We can also connect Algorithm 1 to *differentially private follow-the-regularized-leader* (DP-FTRL) (Kairouz et al., 2021), which uses a tree-based aggregation algorithm to privately release the gradients of the loss function as a prefix sum. This approach provides a competitive privacy/utility tradeoff without relying on privacy amplification or shuffling, which is often not possible in distributed settings. DP-FTRL differs from DP-SGD by adding *correlated* rather than *independent* noise at each iteration; Algorithm 1, in contrast, differs from both by adding *identical* noise at each iteration.

# K Technical Lemmas & Definitions

## K.1 Convex Optimization

We will give a short review of relevant concepts from convex optimization.

*Definition* K.1 ($\ell_2$-Lipschitz continuity). A function $f : \Theta \to \mathbb{R}$ is $L$-Lipschitz w.r.t. the $\ell_2$-norm over $\Theta \subseteq \mathbb{R}^d$ if for all $\theta_1, \theta_2 \in \Theta$, the following holds: $|f(\theta_1) - f(\theta_2)| \leq L||\theta_1 - \theta_2||_2$.

*Definition* K.2 ($\beta$-smoothness). A differentiable function $f : \Theta \to \mathbb{R}$ is $\beta$-smooth over $\Theta \subseteq \mathbb{R}^d$ if its gradient $\nabla f$ is $\beta$-Lipschitz, i.e. $|\nabla f(\theta_1) - \nabla f(\theta_2)| \leq \beta||\theta_1 - \theta_2||_2$ for all $\theta_1, \theta_2 \in \Theta$.

*Definition* K.3 (Strong convexity). A differentiable function $f : \Theta \to \mathbb{R}$ is $\lambda$-strongly convex over $\Theta \subseteq \mathbb{R}^d$ if for all $\theta_1, \theta_2 \in \Theta$: $f(\theta_1) \geq f(\theta_2) + \nabla f(\theta_2)^T(\theta_1 - \theta_2) + \frac{\lambda}{2}||\theta_1 - \theta_2||_2^2$.

## K.2 Differential Privacy

*Definition* K.4 (Privacy loss random variable). Let $\Pr[\mathcal{M}(Z) = \theta]$ denote the probability density of the random variable $\mathcal{M}(Z)$ at output $\theta$. For a fixed pair of neighboring datasets $Z$ and $Z'$, the privacy loss random variable (PLRV) of mechanism $\mathcal{M} : \mathcal{Z} \to \Theta$ is defined as

$$\epsilon_{Z,Z'}(\theta) = \log \frac{\Pr[\mathcal{M}(Z) = \theta]}{\Pr[\mathcal{M}(Z') = \theta]},$$

for the random variable $\theta \sim \mathcal{M}(Z)$.

*Lemma* K.5 (Adaptive composition (RDP) (Mironov, 2017)). *Let $\mathcal{M}_1 : \mathcal{Z} \to \mathcal{R}_1$ be $(\alpha, \epsilon_1)$-RDP and $\mathcal{M}_2 : \mathcal{R}_1 \times \mathcal{Z} \to \mathcal{R}_2$ be $(\alpha, \epsilon_2)$-RDP. Then the mechanism $\mathcal{M} = (m_1, m_2)$, where $m_1 \sim \mathcal{M}_1(Z)$ and $m_2 \sim \mathcal{M}_2(Z, m_1)$, satisfies $(\alpha, \epsilon_1 + \epsilon_2)$-RDP*

*Lemma* K.6 (Change of coordinates). *Consider the map $g : \mathcal{X} \to \mathcal{Y}$ between $\mathcal{X} \subseteq \mathbb{R}^d$ and $\mathcal{Y} \subseteq \mathbb{R}^d$ that transforms $y = g(x)$. Then*

$$\partial y = \left| \det \frac{\partial y}{\partial x} \right| \partial x,$$

*where $\left| \det \frac{\partial y}{\partial x} \right|$ is the absolute value of the determinant of the Jacobian of the map $g$:*

$$\frac{\partial y}{\partial x} = \begin{bmatrix} \partial y_1 / \partial x_1 & \dots & \partial y_1 / \partial x_d \\ \vdots & \ddots & \vdots \\ \partial y_d / \partial x_1 & \dots & \partial y_d / \partial x_d \end{bmatrix}.$$

*Lemma* K.7 (Change of variables for probability density functions.). *Let $g$ be a strictly monotonic function. Then for $y = g(x)$,*

$$p_X(x) = \left| \det \frac{\partial y}{\partial x} \right| p_Y(y).$$

*Lemma* K.8. *Let $A$ be an invertible matrix. Then $\det A^{-1} = \frac{1}{\det A}$.*

*Lemma* K.9 (Maximum Rayleigh quotient). *For any symmetric matrix $A \in \mathbb{R}^{d \times d}$,*

$$\max_{v \in \mathbb{R}^d} \frac{v^T A v}{v^T v} = \lambda_{max},$$

*where $\lambda_{max}$ is the largest eigenvalue of $A$.*

*Lemma* K.10 (Bound on the expected norm of multivariate Gaussian with mean 0.).
Let $x = \mathcal{N}(0, \sigma^2 I_d)$. *Then*

$$\mathbb{E}\left[\,\|x\|_2\,\right] \leq \sigma\sqrt{d}.$$

*Lemma* K.11 (Gaussian MGF). *Let $X \sim \mathcal{N}(\mu, \sigma^2)$ for $\mu \in \mathbb{R}, \sigma \in \mathbb{R}_{>0}$. The moment generating function of order $t$ is then $MGF_X(t) = e^{\mu t + \frac{1}{2}\sigma^2 t^2}$. So for any $t \in \mathbb{R}_{\geq 0}$ and $\sigma_1 < \sigma_2$,*

$$MGF_{X_1}(t) \leq MGF_{X_2}(t),$$

where $X_1 \sim \mathcal{N}(\mu, \sigma_1^2)$ and $X_2 \sim \mathcal{N}(\mu, \sigma_2^2)$.

*Definition* K.12 (Holder's Inequality). Let $X, Y$ be random variables satisfying $\mathbb{E}\left[|X|^p\right] < \infty$, $\mathbb{E}\left[|X|^q\right] < \infty$ for $p > 1$ and $\frac{1}{p} + \frac{1}{q} = 1$. Then

$$\mathbb{E}\left[|XY|\right] \leq \mathbb{E}\left[|X|^p\right]^{\frac{1}{p}} \mathbb{E}\left[|X|^q\right]^{\frac{1}{q}}.$$

*Definition* K.13 (Rényi Divergence). Let $P, Q$ be distributions with probability density functions $P(x), Q(x)$. The Rényi divergence of order $\alpha > 1$ between $P$ and $Q$ is given by

$$D_\alpha(P\|Q) = \frac{1}{\alpha - 1} \log \mathbb{E}_{x \sim Q}\left[\left(\frac{P(x)}{Q(x)}\right)^\alpha\right] = \frac{1}{\alpha - 1} \log \mathbb{E}_{x \sim P}\left[\left(\frac{P(x)}{Q(x)}\right)^{\alpha - 1}\right].$$

*Lemma* K.14 (MGF inequality for identical distributions). *Let $x, y, z \sim \mathcal{N}(0, \sigma^2)$. Then*

$$\mathbb{E}_{x,y}\left[e^{t|z_1|\,|z_2|}\right] \leq \mathbb{E}_z\left[e^{tz^2}\right].$$

*Proof.* Observe that for any $a, b \in \mathbb{R}$, it holds that $2ab \leq a^2 + b^2$. Applying this along with the Cauchy-Schwarz inequality,

$$
\begin{aligned}
\mathbb{E}_{x,y}\left[e^{t|x|\,|y|}\right] &\leq \mathbb{E}\left[e^{\frac{t}{2}x^2} e^{\frac{t}{2}y^2}\right] \\
&\leq \sqrt{\mathbb{E}_x\left[e^{\frac{t}{2}x^2}\right] \mathbb{E}_y\left[e^{\frac{t}{2}y^2}\right]} \\
&= \mathbb{E}_z\left[e^{tz^2}\right].
\end{aligned}
$$

$\square$

