# OpenReview forum: "Improving the Privacy and Practicality of Objective Perturbation for Differentially Private Linear Learners"
_NeurIPS.cc/2023/Conference — NeurIPS 2023 poster_

### Official Review · Reviewer_NU3P · 2023-07-05

**Soundness:** 3 good
**Presentation:** 1 poor
**Contribution:** 2 fair
**Rating:** 3
**Confidence:** 4

**Summary:**

This paper presents a new privacy analysis for the objective perturbation method for learning differentially private linear models. The new analysis makes use of gradient clipping (similar to DP-SGD) and is shown to have a competitive performance with respect to DP-SGD, while exhibiting better behavior for hyperparameter tuning.

**Strengths:**

The problem that this paper sets out to address is very interesting and can potentially have (minor) practical implications.

**Weaknesses:**

My main issue with this paper is the presentation:
Many vague sentences, several terms are not properly defined, confusing notation, and several careless mistakes in mathematical definitions.
I'll be more specific in the following:
1. The first item in the contribution (line 47) states that the PLRV of the objective perturbation mechanism is "almost" Gaussian. The authors failed to elaborate on this point in the draft and it seems to be left to the reader's imagination. The second item (line 53): "We establish even tighter ($\epsilon, \delta$)-DP". Tighter compared to what. The fourth item (line 64) discusses "weakly" differentiable loss functions. What is a weakly differentiable function?
2. The following terms were not defined: $\epsilon(\alpha)$, $L(\theta, Z)$ (line 219),  "privacy profile" (i.e., Lemma  2.6 is very hard to understand without a precise definition of $\delta(\epsilon)$),  $\tilde\theta^P_T$ (in line 229), the set $\mathcal X$ in Theorem 4.2, and also $\hat \theta^P(Z)$ and $\hat \theta^P(Z')$ in Corollary 3.4 (I suspect the authors meant $\hat \theta^P(X)$ and $\hat \theta^P(X')$ as defined in line 190). Finally, the identity $\ell(\theta, z) = \ell(x^T\theta, y)$ in line 141 is very confusing; again I think the authors meant $\ell(\theta, z) = f(x^T\theta, y)$ for some functions $f$.
 3. There are three ways a dataset is referred to in this paper: $Z\in \mathcal Z^n$, $Z\in \mathcal Z$, and $z\in Z^n$!! The notation $\mathbb E [e^{|\mathcal N (0, \sigma^2)|}]$ is not a proper mathematical notation! The authors should have clarified that $\mathcal M(Z)$ denotes both a random variable (i.e., mechanism's output) and its distribution.
4. The definition of R\'enyi divergence in line 108 is WRONG (and the mistake was repeated more than 3 times throughout!!)  Another example: It is stated in line 225 that $\tilde \theta^P = \theta^P + (\tilde \theta - \theta^P)$, which implies $\tilde \theta^P = \tilde \theta$ (which is clearly wrong).

The second weakness is Algorithm 1: First of all, $\gamma$ is one of the inputs but hasn't been used! Moreover, the definition of $\tilde \theta$ is cryptic. This is  the definition:  Solve for $\tilde \theta$  such that $|| \nabla \mathcal L_C^P(\tilde \theta; Z)||_2\leq \tau$. For a given $\tau$, we might have (infinitely many) different such $\tilde \theta$.

The third weakness of this work is the presentation of the main results. It is rather difficult to see the significance of the results compared to the existing privacy analyses of the objective perturbation method. The authors failed to put their results in the context of the known results, so the readers can appreciate them. In particular, it is particularly unclear how Algorithm 1 extends Iyengar et al.'s AMP.

**Questions:**

The authors compared their results for the objective perturbation mechanism with the rather old results for DP-SGD (namely, Bassily et al., 2014). For the convex loss functions, there many more recent privacy analyses for DP-SGD in the literature. For instance, it would be important and insightful if the authors could compare their results with Altschuler and Talwar's result "Privacy of Noisy Stochastic Gradient Descent: More Iterations without More Privacy Loss".


**Limitations:**

Yes

---

> ### Author Rebuttal · Authors · 2023-08-08
>
> Thanks for your feedback! The notation went through a major overhaul during this paper’s lifespan and we obviously didn’t check for consistency as thoroughly as we should have. Our intention was to make this a fun read and we’re sorry to hear it missed the mark for you.
>
> >The first item in the contribution (line 47) states that the PLRV of the objective perturbation mechanism is "almost" Gaussian. The authors failed to elaborate on this point in the draft.
>
> This PLRV is almost Gaussian as it is a shifted half-normal distribution. This is shown explicitly in the appendix (see for example the statement of Lemma E.6). It is also reflected in Thm. 3.1: the log moments are the log moments of a shifted half-normal distribution.
>
> “Almost Gaussian” is also demonstrated in Appendix I, “Distance to Optimality”, which plots the RDP bounds of objective perturbation against those of the Gaussian mechanism and shows that these differences vanish in certain regimes (e.g., large $\alpha$).
>
> We will make sure to state this more clearly in the main paper.
>
> > The second item (line 53): "We establish even tighter $(\epsilon,\delta)$-DP". Tighter compared to what.
>
> Tighter compared to the RDP bound (mentioned in the previous point). We will re-write this to make it clearer.
>
> > The fourth item (line 64) discusses "weakly" differentiable loss functions. What is a weakly differentiable function?
>
> It is a function that has a derivative in the sense of distributions. Weakly differentiable functions are differentiable except for sets of measure zero (the sign function is an example).
>
> But you have a point that the “weakly differentiable” claim might not be appropriate in the introduction, given that our theoretical results don't require it directly. What we wanted to communicate is that Algorithm 1 applies to a broad class of loss functions which only need to satisfy $\beta$-smoothness (including functions which are weakly differentiable).
>
> > The notation $\mathbb{E}...$ is not a proper mathematical notation!
>
> We will change this so that in the bound of Thm 4.1, for example, the last term is given as
> $$
>  \dfrac{1}{\alpha - 1} \log \mathbb{E} \left[e^{(\alpha - 1)\left| X \right|} \right],
>  \quad X \sim \mathcal{N}\left(0, \dfrac{L^2}{\sigma^2}\right).
> $$
>
> > The second weakness is Algorithm 1: First of all, $\gamma$ is one of the inputs but hasn't been used!
>
> Here is again simply a typo: gradient norm threshold $\gamma$ should be $\tau$, we will fix this.
>
> > The third weakness of this work is the presentation of the main results.... In particular, it is particularly unclear how Algorithm 1 extends Iyengar et al.'s AMP.
>
> As we explain in the appendix in Section F, the proof of Kifer et al. (2012) made a certain mistake about independence of random variables, and collapsing $d$-dim Gaussian to a $1$-dim Gaussian. Iyengar et al.'s analysis inherits this same mistake. It is one of our main contributions to provide a correct analysis and also give the AMP version of it.
>
> > Moreover, the definition of $\tilde{\theta}$ is cryptic... For a given $\tau$, we might have (infinitely many) different such $\tilde{\theta}$.
>
> The formulation here is vague by intention... This was also the case in Iyenger et al's AMP paper. Any $\tilde{\theta}$ that satisfies the condition (gradient norm less than $\tau$) will have the rigorous privacy guarantees given by Thm. 4.1. The (blackbox) procedure for finding such $\tilde{\theta}$ is allowed to involve arbitrary data access and tuning.
>
> > The definition of R'enyi divergence in line 108 is WRONG
>
> Thank you for pointing this out, the extra $\log$ is simply a typo and will be fixed.
>
> > It is stated in line 225 that $\tilde{\theta}^P = \theta^P + (\tilde{\theta} - \theta^P)$, which implies $\tilde{\theta}^P = \tilde{\theta}$ (which is clearly wrong).
>
> This is again simply a typo, $\tilde{\theta}$ should be written $\tilde{\theta}^P$.
>
> > The authors compared their results for the objective perturbation mechanism with the rather old results for DP-SGD (namely, Bassily et al., 2014)...it would be important and insightful if the authors could compare their results with Altschuler and Talwar's result "Privacy of Noisy Stochastic Gradient Descent: More Iterations without More Privacy Loss".
>
> First of all, we do not compare to the result of Bassily et al. (2014). As we state on line 280--281, for privacy analysis of DP-SGD we use the analytical moments accountant (Wang et al., 2019) with Poisson sampling (Zhu \& Wang, 2019; Mironov et al., 2019).
>
> The motivation for comparing against DP-SGD is the following. As we write on lines 36--38, empirical evaluations of (Yu et al., 2019; McKenna et al., 2021) suggest that DP-SGD often achieves better utility in practice than objective perturbation and Iyengar et al. (2019) report the opposite. However, as we explain Appendix Section F, Iyengar et al. (2019) have mistakes in their privacy analysis. Therefore, we think it is particularly interesting to compare to DP-SGD with the corrected privacy analysis of the objective perturbation.
>
> Moreover, looking at the analysis of Altschuler and Talwar (2022), (https://arxiv.org/pdf/2205.13710.pdf), we don't think their results are as comparable as they consider the projected noisy-SGD. In their RDP bounds there is always the diameter $D$ of the projection set that would need to be tuned as well.  For the parameter values we have, it is not clear if the convex analysis would improve upon the common (non-convex) DP-SGD RDP analysis. This is reflected in Thm. 1.3 by Altschuler and Talwar (2022), where the bound has the factor $\min \left(T, \frac{D n}{L \eta} \right)$, where $n$ is the size of the dataset, $L$ the Lipscitz constant of the gradients and $\eta$ the learning rate, and where $T$ is the number of training iterations and corresponds the result to the common RDP analysis of the Gaussian mechanism.

---

> > ### Comment · Reviewer_NU3P · 2023-08-16
> > **Thanks for your response.**
> >
> > I thank the authors for the response.
> >
> > My concerns regarding many typos and careless mistakes have been addressed. [The authors should keep in mind that reviewing a manuscript with lots of typos and mistakes leads to frustration.]
> >
> > As for the comparing with DP-SGD, I still think that the authors should have compared their results with more recent analyses of DP-SGD. It is particularly important because we now know that moments accountant (even the analytic moments accountant) tends to overestimate privacy guarantees; see for example Fig 1.b in "Numerical Composition of Differential Privacy" by Gopi et al to see how much gap between moments accountant and the "actual" privacy guarantee. So I still believe the comparison between objective perturbation and DP-SGD presented in this work is not fair and needs further investigation.

---

> > > ### Author Response · Authors · 2023-08-16
> > > **Thanks and some further clarification and perspectives**
> > >
> > > We thank the reviewer for acknowledging our responses and for the follow-up discussion!
> > >
> > > Regarding the
> > > > "(fair) comparison between Objective Perturbation and DP-SGD ..."
> > >
> > > We wanted to add two facts and one opinion which we think could help.
> > >
> > > **Fact 1 Why RDP?** :  We are well aware of the tighter "numerical composition" / privacy profile for DP-SGD.  To make it a fair comparison, we did not use the tighter "numerical composition" for ObjPert (Theorem 3.2) in our experiments either,  but instead used our RDP bound (Theorem 3.1). Using Theorem 3.2 would lead to an improved result for us too. The reason why we compared to the RDP version of the accounting is that the best hyperparameter tuning tool available to date (Steinke and Papernot, 2022) is based on RDP.  It is not compatible with the "numerical accountant" for privacy profile. In addition, RDP is often viewed as a stronger and more elegant relaxation of pure DP that many people in the DP community endorse. It is of value to present results on RDP independent to its implied approximate DP.
> > >
> > > **Fact 2 Purpose of our Experiments**: The reviewer is not wrong that our experimental comparison between DP-SGD and Objective Perturbation is not conclusive in either its breadth (more datasets) or depth (more conditions to explore).   But it was never our intention to demonstrate that Objective Perturbation dominates DP-SGD.  We don't believe this is the case in fact. We merely showed that they achieve comparable performance and sometimes ObjPert is better. The take-home message for practitioners should be that one should treat "objective perturbation" seriously as a strong baseline. We will make this point clearer if the reviewer finds it is currently not.
> > >
> > >
> > > Now let us also add our perspective which may help to convince the reviewer of the value of this paper despite not having a conclusive experimental illustration on DP-SGD vs ObjPert.
> > >
> > > **Opinion:** Instead of ObjPert not comparing fairly to DP-SGD empirically, we would argue the opposite.  It is the DP-SGD literature who should have compared fairly to ObjPert (a long time ago).  "Objective perturbation" is an older and more elegant method compared to DP-SGD.  It is also more computationally efficient.
> > >
> > > Yet, there is nothing a DP practitioner could do. "Objective Perturbation" is simply not on par in its empirical performance because it does not benefit from the newer analysis that applied only to DP-SGD, e.g., RDP for amplification by sampling, privacy profiles of the sampled-Gaussian mechanism and their composition through, e.g., "Numerical Composition of Differential Privacy".
> > >
> > > We don't blame the researchers of existing papers because there is no way they could compare them fairly in experiments.  Comparing to ObjPert using its classical analysis to DP-SGD's modern analysis would be somewhat misleading.  An RDP analysis and a privacy profile analysis for objective perturbation are arguably more difficult to obtain than that for DP-SGD (which is a modular composition or basic mechanisms).  This paper solves the problem by working out both the RDP and privacy profile for objective perturbation. This is our main contribution. Now there is no excuse to not compare to ObjPert. Our results (and code) will make such comparisons easy to do.  In addition, during the course of the project, we fixed a subtle mistake in classical papers on this topic, hence reinforcing the foundation upon which more than ten-year of scientific literature has been developed.
> > >
> > > Considering the technical focus of this paper, we would argue that while a more thorough empirical comparison between DP-SGD and ObjPert could add value, it is *not unreasonable* for us to prefer deferring a more thorough empirical comparison to a longer journal version of this paper or another more empirically-focused paper.
> > >
> > > We hope the above explains our position clearly and we appreciate the reviewer's understanding. Thanks again for your time and discussion!

---

> > > > ### Comment · Reviewer_NU3P · 2023-08-20
> > > > **Thanks for the detailed reponse**
> > > >
> > > > I thank the authors for their detailed response.
> > > >
> > > > I strongly disagree with this argument: " To make it a fair comparison, we did not use the tighter "numerical composition" for ObjPert (Theorem 3.2) in our experiments either, but instead used our RDP bound (Theorem 3.1)."
> > > >
> > > > You're comparing a potentially loose upper bound (the privacy guarantee of ObjPet Thm 3.2) with another loss upper bound (RDP guarantee of SP-SGD). This does NOT make the comparison fair at all! Moreover, the result does not say much about the actual difference between the privacy guarantee of DP-SGD and  that of ObjPert.
> > > >
> > > > As for the comments: "Instead of ObjPert not comparing fairly to DP-SGD empirically, we would argue the opposite. It is the DP-SGD literature who should have compared fairly to ObjPert (a long time ago). "Objective perturbation" is an older and more elegant method compared to DP-SGD. It is also more computationally efficient." and
> > > > I agree that ObjPert is an elegant method (not sure if it is "more elegant" than DP-SGD though!!). But this does not mean that we must throw away recent development of DP-SGD analyses.
> > > >
> > > > And again, I disagree with this comment "Comparing to ObjPert using its classical analysis to DP-SGD's modern analysis would be somewhat misleading."
> > > > Just to emphasize, modern analysis of DP-SGD gives tighter bounds on the $\mathbf{actual}$ privacy guarantees of DP-SGD. So, why do the authors think it is misleading to compare ObjPert with the actual privacy guarantee of DP-SGD?

---

> > > > > ### Author Response · Authors · 2023-08-21
> > > > > **Thanks for the follow-up discussion!**
> > > > >
> > > > > Thanks for the follow-up and for giving us a chance to clarify.
> > > > >
> > > > > The reviewer wrote:
> > > > > > You're comparing a potentially loose upper bound (the privacy guarantee of ObjPet Thm 3.2) with another loss upper bound (RDP guarantee of SP-SGD). This does NOT make the comparison fair at all!
> > > > >
> > > > > We are sorry for the confusion, but we did *not* used our Theorem 3.2 (Privacy profile of ObjPert). We used our Theorem 3.1 (RDP of ObjPert) in the experiments so we are comparing RDP of ObjPert to RDP of DP-SGD.
> > > > >
> > > > > The reason for this choice, as we explained, is that we need to use private hyperparameter selection (Papernot and Steinke, 2022) for DP-SGD, which requires RDP.  Again to be absolutely clear,  the method of Papernot and Steinke works as follows:
> > > > >
> > > > > - It takes the RDP functions of a sequence of mechanisms as input (each with different hyperparameter), then outputs the RDP function for the *meta-mechanism* that returns the argmax according to a utility function, i.e., test accuracy (e.g., on a public holdout dataset).
> > > > >
> > > > > So in order to use this SOTA hyperparameter tuning tool, we need to use the RDP function of a base mechanism (e.g., DP-SGD). An analogous method that takes a tight "privacy profile" analysis (compatible with the Gopi et al or other PLD accountants) as input for DP hyperparameter selection method doesn't exist yet unfortunately.
> > > > >
> > > > > To provide absolute clarity, the two possible choices that we considered for privacy accounting in the experiments are:
> > > > >
> > > > > - **Method 1**. RDP of ObjPert  (Our Theorem 3.1)   vs   RDP of DP-SGD  ( Zhu et al, 2019) + RDP of Hyperparameter Tuning (Papernot and Steinke, 2022).
> > > > >
> > > > > - **Method 2**. Privacy profile of ObjPert (Our Theorem 3.2 )  vs    Privacy profile of DP-SGD (Gopi et al.,  Koskela et al) + Approximate DP of Hyperparameter Tuning ( Liu and Talwar, 2019)
> > > > >
> > > > > We chose **Method 1** in the paper because it is cleaner and more fair for DP-SGD --- it does not suffer from the overhead from a suboptimal private hyperparameter tuning method.  If we are to use the approach in **Method 2**, then one would have to take only one point on the privacy profile (i.e., committing to one fixed $\epsilon,\delta$ pair), then pass it into Liu and Talwar's bound (Theorem 3.4 of https://arxiv.org/pdf/1811.07971.pdf).  From our experience, **Method 2** for DP-SGD will provide *worse* approximate-DP parameters compared to that of **Method 1**  for DP-SGD in the meaningful parameter ranges (small $\delta$, moderate $\epsilon$).  Meanwhile, ObjPert will enjoy stronger privacy parameters under **Method 2** as Figure 1 in our paper clearly demonstrates.
> > > > >
> > > > > We are grateful to the reviewer for the discussion so far. It shows that the above isn't clear to readers and we should've explained these in the initial submission. Moreover, we now believe it is important to explicitly report results under **Method 2** too.  So that is what we will do in our revision. Hopefully, this addresses the reviewer's concern.  The experiments are underway, if we are able to code it up fast enough before the discussion period expires, we will report back with concrete numbers.
> > > > >
> > > > > The reviewer also wrote
> > > > > > ... modern analysis of DP-SGD gives tighter bounds on the
> > > > >  privacy guarantees of DP-SGD. So, why do the authors think it is misleading to compare ObjPert with the actual privacy guarantee of DP-SGD?
> > > > >
> > > > > We agree it is important to compare actual privacy guarantees. The issue is that the actual privacy guarantees for ObjPert were not worked out.  This paper gets very close in achieving comparable modern privacy accounting for ObjPert (See Figure 1 and Appendix I, note that GM is a lower bound for ObjPert).
> > > > >
> > > > > As the reviewer pointed out, the improvements from Bassily et al (2014) to Abadi et al. (2016), Zhu et al. (2019) then to Gopi et al. (2021) are substantial. We believe DP-SGD became practical because of these modern analyses. If the reviewer also agrees to this statement and the fact that --- prior to this paper --- the analysis of ObjPert is still analogous to that of Bassily et al. (2014), we believe the reviewer would agree to the value of our paper in coming up with the counterparts of Zhu et al. (2019) then to Gopi et al. (2021) for ObjPert.
> > > > >
> > > > > Thank you again for engaging in the discussion with us. We hope our response helps!

---

> > > > > > ### Author Response · Authors · 2023-08-21
> > > > > >
> > > > > > In the table below, we've compared the $(\epsilon,\delta)$-bounds obtained using Thm. 6 by Papernot and Steinke (2022)
> > > > > > that takes as an input the RDP parameters of DP-SGD and Thm. 3.5 by Liu and Talwar (2019)
> > > > > > that takes as an input the $(\epsilon,\delta)$-bounds of DP-SGD
> > > > > > (computed using the PRV accountant by Gopi et al., 2021).
> > > > > >
> > > > > > $\epsilon_{M1}$ is the privacy parameter for Method 1 (RDP of DP-SGD + Papernot & Steinke hyperparameter tuning) and $\epsilon_{M2}$ is the privacy parameter for Method 2 (PRV accountant + Liu & Talwar hyperparameter tuning). Note that some of the deltas are a little large due to numerical constraints imposed by Thm 3.5 of Liu & Talwar. The parameters in both cases are set such that
> > > > > > in expectation the tuning algorithm evaluates 15 hyperparameter values.
> > > > > >
> > > > > > The subsampling ratio for the underlying DP-SGD is the same as in our experiments for the Adult dataset.
> > > > > >
> > > > > > | $\\delta$    | $\\epsilon_{M1}$ | $\\epsilon_{M2}$ |
> > > > > > | -------- | ------- | ------- |
> > > > > > | $2.5e^{-4}$ |  2.004 | 3.878
> > > > > > | $9.23e^{-4}$ | 1.88 | 3.62
> > > > > > | 0.002  | 1.795    | 3.457
> > > > > > | 0.004 | 1.695     | 3.288
> > > > > > | 0.009    | 1.587    | 3.11
> > > > > > | 0.04 | 1.326 | 2.726
> > > > > >
> > > > > > For dishonest DP-SGD, we do get some marginal improvements by using the PRV accountant. But we hope that this demonstrates that when accounting for the cost of hyperparameter tuning, using numerical composition with the Liu & Talwar bounds is in fact suboptimal compared to the RDP bounds!

---

### Official Review · Reviewer_eK3K · 2023-07-06

**Soundness:** 4 excellent
**Presentation:** 4 excellent
**Contribution:** 3 good
**Rating:** 7
**Confidence:** 5

**Summary:**

The paper revisited a differentially private learning algorithm: objective perturbation. The authors gave an RDP analysis of objective perturbation and extended AMP to losses with unbounded gradients using clipped loss function.

**Strengths:**

The main contributions of the paper are (1) RDP and HK analysis of objective perturbation; (2) an improved version of AMP which extends to losses with unbounded gradients.

I really enjoyed reading the manuscript. The authors did a fantastic job telling the story why objective perturbation should regain attention from the community.

The analysis of HK and RDP bound, despite not challenging, are good contribution. Another contribution is that the authors found an unacknowledged subtlety in the analysis of objective perturbation and pointed out GLM assumption solved the issue.

The extended AMP is a direct combination of clipped loss function and AMP. Again, not too technically challenging, but solid contribution.

Another contribution is that the authors showed that objective perturbation is computationally more efficient than DP-SGD.

**Weaknesses:**

1. Compared to the pretty theory and story-telling, the evaluation is somewhat weak. First, objective perturbation does not consistently show better performance than honest DP-SGD. Second, even in cases where objective perturbation wins, the winning is marginal. Third, objective perturbation is only evaluated on 2 toy datasets. More realistic datasets are preferred.
2. The 3rd bullet point in the contribution list emphasized that there is neglected subtlety in the analysis of objective perturbation which might lead to dimensional dependence of PLRV and the paper fixed the issue without touching the topic in detail except a pointer to appendix in remark 3.5. I think this contribution deserves more space in the main text.
3. typo: line 306 "boudning" -> "bounding"

**Questions:**

Please add more experiments if possible

**Limitations:**

Yes

---

> ### Author Rebuttal · Authors · 2023-08-09
>
>
> Thank you for the review, we are happy to hear you enjoyed reading the paper!
>
> > The 3rd bullet point in the contribution list emphasized that there is neglected subtlety in the analysis of objective perturbation which might lead to dimensional dependence of PLRV and the paper fixed the issue without touching the topic in detail except a pointer to appendix in remark 3.5. I think this contribution deserves more space in the main text.
>
> Thank you for the feedback, we will give more space for this in the main text.
>
> On the same subject, we also didn’t emphasize that Iyenger et al’s 2019 analysis inherited the mistake from the classical analysis. One of our other contributions was to provide a new privacy analysis of AMP that fixes this. We stated this only informally in Appendix H.2; the next revision of the paper will formalize this argument!
>
> >  the evaluation is somewhat weak.. objective perturbation does not consistently show better performance than honest DP-SGD... even in cases where objective perturbation wins, the winning is marginal... More realistic datasets are preferred...
>
> The experimental evaluation is definitely a weakness! We are working on scaling up the experiments and the complexity of the datasets. In the meantime, we also plan to look more closely into the small $\epsilon$ setting (e.g., $\epsilon = 0.05, 0.01$) where we suspect that the effect of hyperparameter tuning is more pronounced.
>
> > typo: line 306 "boudning" -> "bounding"
>
> Thanks, we will fix this.

---

> > ### Comment · Reviewer_eK3K · 2023-08-16
> > **thanks for the rebuttal**
> >
> > Thanks for the rebuttal. I will keep my score.

---

### Official Review · Reviewer_RDzn · 2023-07-11

**Soundness:** 3 good
**Presentation:** 3 good
**Contribution:** 3 good
**Rating:** 5
**Confidence:** 2

**Summary:**

The paper revisits objective perturbation as a viable alternative to DP-SGD with tighter analysis and illustrating it competitiveness on a few problems.

**Strengths:**

This paper explores the interesting direction of fairly comparing objective perturbation and DP-SGD specifically. The paper has multiple strengths

- Paper presents a novel analysis of objective perturbation using RDP, and extending it to an even tighter bound
- It also empirically verifies that its competitive against DP-SGD, especially on lower epsilons, while having the advantage of "free" hyperparam tuning
- Introduces an algorithm to expand the applicability of objective perturbation to a broad range of losses

**Weaknesses:**

Here are a few weaknesses:

- Empirical evaluation of this method can be improved, I had following questions in this regard
  - DP-SGD is known to work best with large batch sizes, I would suggest increasing batch size for for the baselines -- even better is to have a plot with varying batch sizes. Similar argument can made for number of iterations as well.
  - Even though there is a good reason to use L-BFGS with Alg 1, including an ablation when using Adam with Alg 1 would be useful as a comparison.
-  While not a major one for publication but a still a weakness: the function class that analysis applies to for objective perturbation is limited compared to DP-SGD.
- Another weakness is Alg 1's either matches or performs worse against DP-SGD on epsilons > 0.1

**Questions:**

Please see "weaknesses" section

**Limitations:**

Yes

---

> ### Author Rebuttal · Authors · 2023-08-09
>
> > DP-SGD is known to work best with large batch sizes, I would suggest increasing batch size for for the baselines -- even better is to have a plot with varying batch sizes. Similar argument can made for number of iterations as well.
>
> The batch size and the number of iterations were simply reasonable sounding first choices (see e.g. the experiments by Abadi et al., 2016, Deep learning with differential privacy). If we tune the batch size and number of iterations in addition to the learning rate as well, it will likely worsen the privacy utility tradeoff for honestly tuned DP-SGD, as we need to sample much more hyperparameters (from a product grid, e.g.) and the privacy cost of the hyperparameter tuning algorithm by Steinke and Papernot (2022) increases.
>
> > Even though there is a good reason to use L-BFGS with Alg 1, including an ablation when using Adam with Alg 1 would be useful as a comparison.
>
> Thanks for the suggestion! We don't *think* that using Adam instead of L-BFGS in Alg. 1 would affect the utility of the final result, as we are simply minimizing the (convex) perturbed loss function until the stopping criterion (determined by the tolerance parameter $\tau$) is fulfilled. But it would be good to take a closer look!
>
> > Another weakness is Alg 1's either matches or performs worse against DP-SGD on epsilons > 0.1
>
> This is likely due to the fact that the model accuracies in these examples do not really improve when going beyond $\epsilon=1.0$ (notice that Alg. 1 is never significantly worse). We agree that improvements on DP-SGD seem to be biggest for small $\epsilon$'s in these examples. We will add results for small $\epsilon$-values.

---

> > ### Comment · Reviewer_RDzn · 2023-08-18
> > **Thank you**
> >
> > Thanks for the rebuttal, I will keep my score based on the relative empirical performance of this method

---

### Official Review · Reviewer_udpp · 2023-07-21

**Soundness:** 4 excellent
**Presentation:** 2 fair
**Contribution:** 4 excellent
**Rating:** 6
**Confidence:** 4

**Summary:**

This paper derives new privacy analysis of the objective perturbation mechanism. Novel bounds are given for Rényi differential privacy and approximate differential privacy, that are tighter than existing bounds. A practical variant of objective perturbation (where the problem does not have to be solved exactly) is proposed, and evaluated numerically.


**Strengths:**

1. The RDP analysis of objective perturbation is an interesting and original contribution, and the proposed analyses improve over existing results on the privacy of objective perturbation.
2. A practical variant of the method is proposed, with a convergence criterion that allows tuning the hyperparameter of the underlying optimization sub-routine for free (in terms of privacy).
3. Empirical results are quite convincing, matching the utility of DP-SGD without requiring a heavy tuning procedure.


**Weaknesses:**

1. The proposed method is compared with DP-SGD for non-smooth functions [1], which is a different setting. In the smooth setting, DP-SGD and its variants converge faster, which makes the argument that the proposed Objective Perturbation method is more computationally efficient vacuous. (In general, I don't find the discussion on computational cost is so useful, and it could be mostly removed to incorporate more discussion on other results.)
2. Some literature on differentially private generalized models is missing. In particular, [2] could be a good baseline and is not mentioned in the paper.
3. No closed-form on the value of $\sigma$ required to obtain a given RDP or DP guarantee is provided. This makes the comparison with existing results a bit difficult.

In terms of structure, the paper is not too convincing: a large part of the contributions is deferred to the appendix, and references to appendix in the main text are vague. In general, results are presented without much discussions. In particular, Theorem 3.1 (central result of the paper) is presented without details on the comparison with Theorem 2.8; and Figure 1 is not discussed (and lacks a bit of details for proper understanding). It also seems that a title is missing between lines 184 and 185

Some statements are also not very clear: the regularization $r(\theta)$ disappears in the definition of $J^P$ (page 4); $\hat \theta$ in Corollary 3.4 seems to be the one from Example 3.3 but it is not stated; the problem considered in Figure 1 is not stated.

[1] Private empirical risk minimization: Efficient algorithms and tight error bounds, Raef Bassily et al., 2014.
[2] Differentially Private Generalized Linear Models Revisited, Raman Arora et al., 2022.


**Questions:**

1. Theorem 2.8 is stated for a fixed $\epsilon$, while other theorems fix $\sigma$ and express the privacy parameters ($\alpha, \epsilon$ or $\delta$) as a function of $\sigma$. What is the reason? It seems that unifying this would significantly help the discussion (e.g. by stating Theorem 2.8 for fixed value of $\sigma$). Conversely, is it possible to give a closed-form for $\sigma$ in Theorems 3.1 and 3.2?
2. In Figure 1, it seems that the gap between Kifer's results and RDP/HS for ObjPert reduces as $\sigma$ increases. Is it the case, and is it coherent with the theory? Are there some regimes where both analyses give equivalent results?
3. Why is the comparison with DP-SGD with "honest"-tuning so important? It seems that empirical results are competitive even with the tuned DP-SGD, which hints that tuning is not so fundamental in the comparison between the two mechanisms.
4. It is a bit weird to talk about "DP-SGD with ADAM optimizer" (e.g., this variant is not mentioned in [1]): what does it means? Why not compare with the usual DP-SGD?
5. In general, SGD is known not to be the best optimizer for GLMs. Another approach, closely related to Objective Perturbation, that could be better, is *Output* Perturbation. Why not compare with Output Perturbation? Can the ideas in this paper be used to improve Output Perturbation?

[1] Private empirical risk minimization: Efficient algorithms and tight error bounds, Raef Bassily et al., 2014.


**Limitations:**

The paper is mostly limited to GLMs, which is clearly stated by the authors. The baselines considered for comparing computational cost are a bit weak (e.g., DP-SGD from [1] does not assume smoothness), which reduces the strength of the argument on the computational cost.

---

> ### Author Rebuttal · Authors · 2023-08-08
>
> > The proposed method is compared with DP-SGD for non-smooth functions [1], which is a different setting. In the smooth setting, DP-SGD and its variants converge faster, which makes the argument that the proposed Objective Perturbation method is more computationally efficient vacuous. (In general, I don't find the discussion on computational cost is so useful, and it could be mostly removed to incorporate more discussion on other results.)
>
> We are aware of some literature showing that DP-SGD runs in linear time if the objective is smooth, e.g.:
>
> Wang, Di, Minwei Ye, and Jinhui Xu. "Differentially private empirical risk minimization revisited: Faster and more general." Advances in Neural Information Processing Systems 30 (2017).
>
> This paper presents a DP-version of SVRG, with a claim that DP-SVRG achieves nearly linear time complexity in getting to the information-theoretic limit. But the main result missed the nonlinear regime of amplification by sampling, i.e., when $\\epsilon$ is large, you do not get an amplification. For that reason, the linear-time complexity of DP-SVRG only applies to cases when $\\epsilon$ is as small as $1/n$. (This was not stated in their Theorem 4.3 and 4.4, but looking at the proof (Page 13 of [WYX'17]) shows that an additional condition of small $\\epsilon$ is needed.)
>
> The above confusion might have been popularized by the (very nice) paper by Kulkarni, Lee, and Liu (2021) who focused on the non-smooth case, but *incorrectly* cited [WYX'17] for obtaining a nearly linear time for DP-ERM with smooth/convex losses.
>
> There is also this paper:
>
> Feldman et al. Private stochastic convex optimization: optimal rates in linear time. STOC 2020.
>
> But they study only the DP-SCO setting which is not comparable to our DP-ERM setting.
>
> Please let us know if we’ve missed any literature! We are ready to argue that smoothness doesn't actually help DP-SGD to get faster than O(n^2) :)
>
> > Some literature on differentially private generalized models is missing. In particular, [2] could be a good baseline and is not mentioned in the paper.
>
> Thanks for this reference! It does indeed look very useful and relevant.
>
> > Theorem 2.8 is stated for a fixed $\\epsilon$, while other theorems fix $\\sigma$ and express the privacy parameters ($\\alpha$, $\\epsilon$ or $\\delta$) as a function of $\\sigma$. What is the reason? It seems that unifying this would significantly help the discussion (e.g. by stating Theorem 2.8 for fixed value of $\\sigma$). Conversely, is it possible to give a closed-form for $\\sigma$ in Theorems 3.1 and 3.2?
>
> Stating $\\epsilon$ as a function $\\sigma$ in Thm. 2.8 is straightforward (it's already in the proof from Appendix C.1 of Kifer et al., 2012 ), in our notation, the expression is $\epsilon = \frac{L^2}{\sigma^2} + \frac{L}{\sigma}\sqrt{8\log (2/\delta)}$. We can state the bound for $\\epsilon$ instead as you suggested.
> Comparing different bounds analytically is quite difficult, however we remark that the techniques behind Thm. 2.8, 3.1 and 3.2 are quite different (analytical bounds, RDP bounds, and $(\\epsilon,\\delta)$-bounds directly using the hockey-stick divergence, respectively) and this shows up in the numerically evaluated bounds of Figure 1 as well.
>
> > In Figure 1, it seems that the gap between Kifer's results and RDP/HS for ObjPert reduces as $\\sigma$
>  increases. Is it the case, and is it coherent with the theory? Are there some regimes where both analyses give equivalent results?
>
> This is a great question --- we likely can't give a really satisfying answer without digging into the proof details in Appendix C of the Kifer et al (2012) paper.  Our original inspiration for pursuing an RDP/HS analysis for ObjPert is that we noticed that some of the inequalities in the Kifer et al proofs looked pretty loose; it's possible that they are tighter in certain regimes.
>
> > It is a bit weird to talk about "DP-SGD with ADAM optimizer" (e.g., this variant is not mentioned in [1]): what does it means? Why not compare with the usual DP-SGD?
>
> We wanted to emphasize that our choice of optimizer (Adam) was a hyperparameter. It is the usual DP-SGD mechanism for releasing stochastic gradient, then passed into the torch.optim.Adam instead of  torch.optim.SGD. This is the version implemented in Opacus.
>
> > Why is the comparison with DP-SGD with "honest"-tuning so important? It seems that empirical results are competitive even with the tuned DP-SGD, which hints that tuning is not so fundamental in the comparison between the two mechanisms.
>
> In our experiments, tuning seems to be more important in the small $\\epsilon$ regime, where we speculate that DP-SGD has no privacy budget to spare. (Imagine the effect of taking \\$100 from a grad student vs a CEO. DP-SGD with small $\\epsilon$ is the grad student and DP-SGD with large $\\epsilon$ is the CEO.) We plan to look more closely into this small $\\epsilon$ regime.
>
> > In general, SGD is known not to be the best optimizer for GLMs. Another approach, closely related to Objective Perturbation, that could be better, is Output Perturbation. Why not compare with Output Perturbation? Can the ideas in this paper be used to improve Output Perturbation?
>
> If we had to guess: our analysis techniques would likely make no difference, since the bound for output perturbation (at least, the Gaussian mechanism) is already tight. But the privacy analysis of output perturbation is based on the maximum gradient norm, so our gradient clipping technique could likely apply there. Thanks for this interesting line of thought!

---

> > ### Comment · Reviewer_udpp · 2023-08-13
> >
> > I thank the authors for their detailed response. I appreciate that authors consider adding results in the high-privacy regime (I find it particularly nice that the results get better for smaller $\epsilon$), although I second Reviewer MKLH's comments that more experiments are needed.
> >
> > Although not too critical, I believe that results could be discussed in more depth: more details could be given on the motivation of refining Kifer et al., 2012's results (e.g. what are the inequalities that are not tight, how the proposed analysis is better, and in which regimes do we observe the most gains). It is a nice contribution to remark and fix the errors in their proof and of AMP. Nonetheless, I would appreciate if some intuition on Theorem 3.1's results was given. (By the way, isn't it possible to give a closed-form expression for $\mathbb{E} (\exp(\mathcal{N}(0, L^2/\sigma^2))$?)
> >
> > As for the convergence rate of DP-SGD, provided gradient's variance at optimum is small enough, I believe the rate of DP-SGD with constant step size for convex and smooth functions (which seems to be the setting of the experiments) is $O(1/T + T/n^2\epsilon^2)$, which is minimal when $T=O(n)$. Or am I missing something? In any case, this discussion is not critical for the relevance of the paper anyway.

---

> > > ### Author Response · Authors · 2023-08-16
> > >
> > > > Although not too critical, I believe that results could be discussed in more depth: more details could be given on the motivation of refining Kifer et al., 2012's results
> > >
> > > This is a good suggestion; we’ll try to explain more concretely. The big idea is that the Kifer et al analysis is from 2012 and 11 years later, we have more tools for tighter privacy analysis / accounting. In particular, RDP and privacy profiles are a known vehicle for concrete improvements over classical DP calculations.
> > >
> > > One way to see that the Kifer et al approach isn't optimal: in the proof of their Lemma 14, they are composing an $(\epsilon, 0)$-DP bound (restricting the output space to the set GOOD) with a $(0, \delta)$-DP bound (restricting the output space to the set NOT GOOD). Composition with RDP (then converting to ($\epsilon, \delta$)-DP at the end) is known to be tighter than composing with ($\epsilon, \delta$)-DP.
> > >
> > > We can also describe the difference between the Kifer et al. bounds and privacy profiles. Consider the privacy loss random variable of a mechanism $\mathcal{M}$ for neighboring datasets $Z$ and $Z'$,
> > > $$
> > > \\epsilon_{Z,Z’} = \\log \\frac{ \\mathrm{Pr}[\mathcal{M}(Z) = \\theta ] }{ \\mathrm{Pr}[\\mathcal{M}(Z’) = \\theta ] }
> > > $$
> > >
> > > for the random variable $\\theta \sim \mathcal{M}(Z)$. The $(\epsilon, \delta)$-privacy profile is obtained from the fact that $\mathcal{M}$ is $(\epsilon, \delta)$-DP iff, for all neighbouring $Z$ and $Z'$,
> > > $$
> > > \\mathrm{Pr}[ \\epsilon_{Z,Z’} \\geq \\epsilon ] - e^{ \\epsilon} \\mathrm{Pr}[  \\epsilon_{Z,Z’} \\leq -\\epsilon ] \leq \\delta.
> > > $$
> > >
> > > Instead, Kifer et al. consider the tail bound
> > > $$
> > > \\mathrm{Pr}[ \\epsilon_{Z,Z’} \\geq \\epsilon ] \\leq \\delta
> > > $$
> > >
> > > which zeroes out the negative term and therefore always leads to an upper bound for the privacy profile.
> > >
> > > > ...in which regimes do we observe the most gains
> > >
> > > We will add a more thorough discussion of this in the paper.
> > >
> > > For now, here is a small example: for $\\delta=1e^{-5}$ and $\\epsilon = (0.1, 1, 8)$, we calibrated the parameters $\\sigma, \\lambda$ according to our hockey-stick bound. For the same parameters the Kifer et al bound guarantees $\\epsilon = (0.23,1.91,13.1)$. So these numerical evaluations suggest that the privacy gains are largest for small $\\epsilon$ (though roughly a factor of 2 improvement across the board).
> > >
> > > > By the way, isn't it possible to give a closed-form expression for $\\mathbf{E}( \\exp(\\mathcal{N}(0,L^2/\\sigma^2)))$?
> > >
> > > We stated our bounds this way to make them easier to read, but yes, our RDP bound can be expressed analytically. We believe the reviewer is referring to the expectation $\\mathbf{E} [e^{(\\alpha-1) |X|}]$, $X \sim \\mathcal{N}(0,\\tfrac{L^2}{\\sigma^2})$, appearing in Thm. 3.1.
> > > If $X \\sim \\mathcal{N}(\\mu,\\sigma^2)$, then the moment generating function of the folded normal distribution is
> > > $$
> > > \\mathbf{E} [ e^{ \alpha |X|}] = e^{ \\frac{\\sigma^2 \\alpha^2}{2} + \\mu \\alpha}   \\Phi\left( \\frac{\\mu}{\\sigma} + \\sigma \\alpha \\right) + e^{ \\frac{\\sigma^2 \\alpha^2}{2} - \\mu \\alpha}   \\Phi \\left( - \\frac{\\mu}{\\sigma} + \\sigma \\alpha \\right),
> > > $$
> > > where $\Phi$ denotes the normal CDF. We will add this to the paper.

---

> > > > ### Author Response · Authors · 2023-08-16
> > > > **Bonus round: convergence rate of DP-SGD**
> > > >
> > > > > As for the convergence rate of DP-SGD...
> > > >
> > > > Let us clearly illustrate how the $O(n^2)$ complexity comes about!
> > > >
> > > > The function to optimize is $f(\\theta) = \\sum_i \\ell_i(\\theta)$   (notice that this is the aggregate not the average version of ERM).  We assume each $\\ell_i$ is $L$-Lipschitz and $\beta$-smooth.  So the overall objective is $n\beta$-smooth.   By the standard convergence bound of SGD (e.g., from Ghadimi and Lan) with an unbiased stochastic gradient oracle satisfying a variance bound $\sigma^2$, the convergence result after $T$ iterations (assuming the standard condition $\eta \\leq \\frac{1}{n\\beta}$ to avoid non-convergence even with a full-batch gradient)  is
> > > > $$
> > > > \\mathbb{E}[ f(\\hat{\\theta})  - f^*] \\leq \\frac{\\|\\theta^*\\|^2}{T\eta}  +  \\sigma^2 \\eta.
> > > > $$
> > > >
> > > > So it boils down to the choice of $\\eta$, $T$, and $\\sigma$.  We will start with $\\sigma$, then choose $\\eta$ optimally then discuss on the choices of $T$.
> > > > To begin the unbiased gradient estimator is
> > > > $$
> > > > g(\\theta) =  \\frac{1}{\\gamma} \\left(\\sum_{i\\in \\text{Minibatch}} \\nabla \\ell_i(\theta)  +  \\text{DP Noise}\\right)
> > > > $$
> > > >
> > > > So the variance term $\\sigma^2$ decomposes into the noise from DP and the noise from stochastic sampling. The part from DP,  in order to satisfy $\\rho$-tCDP (similar to $(\\epsilon,\\delta)$-DP with small $\\delta$ and $\\epsilon \\asymp \\sqrt{\\rho}$), is $\\frac{dTL^2}{2\\rho\\gamma^2}$.    The part from sampling with probability $\\gamma$ is $\\frac{\\gamma nL^2}{\\gamma^2} = \\frac{nL^2}{\\gamma}$.
> > > >
> > > > Substituting to the convergence bound, we get
> > > > $$
> > > > \\mathbb{E}[ f(\\hat{\\theta})  - f^*] \\leq \\frac{\\|\\theta^*\\|^2}{T\eta}  +  \\left(  \\frac{d TL^2}{2\\gamma^2 \\rho} + \\frac{n L^2}{\\gamma} \\right) \\eta \\asymp  \\frac{n\beta\|x_1-x^*\|^2}{ T} +  \\sqrt{\\frac{dL^2\\|\\theta^*\\|^2}{\rho}} + \\sqrt{ \\frac{\\|\\theta^*\\|^2 n L^2}{ T\gamma} }
> > > > $$
> > > >
> > > > where the last step is obtained by choosing the learning rate $\\eta$ optimally.  Note that the middle term is the minimax optimal rate for DP-ERM for convex and Lispchitz loss. We need to choose $T$ to be sufficiently large so the middle term dominates --- thus we achieve the optimal rate.
> > > >
> > > > In order for the third term to be on the same order as the middle term, we need $T > O(n/\\gamma)$ (omitting dependence on other quantities). Note that each iteration takes on average $\\gamma n$ incremental gradient computations, so the total computation needs to be larger than $ T \\times \\gamma n \\asymp n^2 )$.
> > > >
> > > > Back to what you wrote:
> > > >
> > > > > I believe the rate of DP-SGD with constant step size for convex and smooth functions (which seems to be the setting of the experiments) is $O(1/T + T / n^2 \\epsilon^2 )$, which is minimal when $T = O(n)$.  Or am I missing something?
> > > >
> > > > We believe the main differences are:
> > > > - you are using the **average** objective rather than the **sum** objective for DP-ERM (a factor of $1/n$ multiplied to the optimal rate)
> > > > - The calculations **missed the variance from sampling the minibatch**.   If you are talking about full-batch gradient, then $O(n)$ iterations is correct but then the computation is still $O(n^2)$ since each iteration costs $n$ incremental gradient oracle calls.
> > > >
> > > > We hope the above is clarifying and fully addressed your concern. (We also hope we didn't overwhelm you with information.)

---

### Official Review · Reviewer_MKLH · 2023-07-28

**Soundness:** 4 excellent
**Presentation:** 4 excellent
**Contribution:** 3 good
**Rating:** 7
**Confidence:** 4

**Summary:**

The paper considers a practical perspective on privacy for unconstrained convex generalized linear problems.  While DP-SGD has been the workhorse for differential privacy in deep learning and prediction/classification problems, the authors suggest that objective perturbation might be more suitable for simpler models.  (Also to provide a more 'honest' privacy mechanism.)
The authors provide privacy bounds for objective perturbation based on Renyi DP.  Using the hockey-stick divergence, the authors provide a tighter $(\epsilon,\delta)$-DP bound. The authors also introduce computational tools to avoid the necessity of bounded gradient.

The paper is clear and enjoyable to read.  The authors perform a good analysis of DP in linear settings, and provide usable practical tools.  While the numerical results do not demonstrate a significant improvement with the proposed algorithm, the importance to using objective perturbation or looking beyond DP-SGD in practice is a valuable analysis.

**Strengths:**

The paper is technically sound - the authors carry out a good analysis and propose an objective perturbation method using various frameworks such Renyi DP and the hockey-stick divergence.
The paper makes a meaningful contribution to the practical applicability of DP.

**Weaknesses:**

A gradient clipping parameter is used in the proposed algorithm - authors should explain why this does not lead a more 'dishonest' privacy mechanism.

The paragraph that explains the tuning of privacy parameter of objective perturbation (starting line 287, page 9) is not very clear.  How is it that the tuning is data-independent?

The results on the Adult dataset show that the proposed algorithm performs close to dishonest DP-SGD, but actually the accuracy values of all three methods are quite close.  For the synthetic dataset on the other hand, the algorithm doesn't show an improved performance, even over the honest method. Why is that?  Experiments on more datasets would be necessary to support the importance of objective perturbation.

**Questions:**

A gradient clipping parameter is used in the proposed algorithm - authors should explain why this does not lead a more 'dishonest' privacy mechanism.

The paragraph that explains the tuning of privacy parameter of objective perturbation (starting line 287, page 9) is not very clear.  How is it that the tuning is data-independent?

The results on the Adult dataset show that the proposed algorithm performs close to dishonest DP-SGD, but actually the accuracy values of all three methods are quite close.  For the synthetic dataset on the other hand, the algorithm doesn't show an improved performance, even over the honest method. Why is that?  Experiments on more datasets would be necessary to support the importance of objective perturbation.

**Limitations:**

The authors have not addressed limitations.

---

> ### Author Rebuttal · Authors · 2023-08-08
>
> > A gradient clipping parameter is used in the proposed algorithm - authors should explain why this does not lead a more 'dishonest' privacy mechanism.
>
> The clipping parameter $C$  is indeed a parameter that affects the privacy guarantees, and tuning it would have an additional privacy cost. However, for logistic regression (with a bias term), we have a natural choice $C = \\sqrt{2}$  for the clipping constant. Notice also that we are free to tune the rest of the hyperparameters of the optimizer (learning rate, batch size) "for free”.
>
> We wanted to fix $C$ across both algorithms so that we could isolate the effect of tuning optimization-related parameters (batch size, number of iterations) which don’t affect the privacy guarantee of objective perturbation / Algorithm 1, but do affect the privacy guarantee of DP-SGD. The clipping parameter $C$ would affect both algorithm’s privacy guarantees!
>
> > The paragraph that explains the tuning of privacy parameter of objective perturbation (starting line 287, page 9) is not very clear. How is it that the tuning is data-independent?
>
> Likely it will be clearer to state this as pseudo-code in the appendix, rather than trying to explain in words! What we meant is that we can choose the noise scale $\\sigma$ and the regularization strength $\\lambda$ without looking at the data.
>
> The privacy guarantees of objective perturbation depend on $\\lambda$ and $\\sigma$. We don’t want to add too much bias (large $\\lambda$) because it will hurt utility, but if we set $\\lambda$ too small then there is a risk that our $\\sigma$ would need to become unreasonably large (and also hurt utility).
>
> How do we quantify when $\\sigma$ is “too large”? We use the Gaussian mechanism as a reference point: the noise scale for objective perturbation shouldn’t be too much larger than the noise scale for the Gaussian mechanism. Let’s say that the Gaussian mechanism with noise scale $\\sigma_G$ satisfies ($\\epsilon, \\delta$)-DP, then we want our $\sigma$ for ($\\epsilon, \\delta$)-DP objective perturbation to satisfy $\\sigma \\leq c \sigma_G$ for some small constant factor $c$.
>
> So our strategy is to find the smallest possible $\\lambda$ such that $\\sigma$ isn’t "too large”. We can accomplish this using fixed parameters (e.g., $\\epsilon, \\delta, \\sigma_G$) that are independent of the data!
>
> > The results on the Adult dataset show that the proposed algorithm performs close to dishonest DP-SGD, but actually the accuracy values of all three methods are quite close. For the synthetic dataset on the other hand, the algorithm doesn't show an improved performance, even over the honest method. Why is that?
>
> We agree that we need to scale up the experiments! Some issues there are
>
> 1. *Poor choice of datasets*. In hindsight, the synthetic and Adult datasets are too simple to properly showcase the benefits of Algorithm 1 (and the disadvantages of DP-SGD). In this case tuning likely doesn’t matter too much as an extensive search isn’t necessary to find a good set of hyperparameters.
> 2. *Searching the wrong regimes*. Our choice of $\\epsilon$ included some redundancies: Going from $\\epsilon = 1$ to $\\epsilon = 8$ didn’t make much difference in accuracy neither for DP-SGD nor Algorithm 1. Likely this is because the datasets were already “easy” and so more privacy budget after $\\epsilon = 1$ gave diminishing returns.  On the other hand, for both the synthetic and Adult datasets we saw the largest improvements for $\\epsilon= 0.1$ — our response to Reviewer udpp includes our intuition on why this is the case. So we plan to include more results on the small $\\epsilon$ regime.

---

> > ### Comment · Reviewer_MKLH · 2023-08-17
> >
> > I thank the authors for their detailed explanation and response to my review.  My concerns about the setting and analysis have been addressed.
> > My concerns about the experimental evaluations remain however.  As the authors themselves state, the choice of datasets is not appropriate to show what the authors would like to show.  I would strongly suggest to the authors to run their experiments on more suitable datasets for the camera-ready version, if the paper is accepted.  The authors really do a disservice to themselves with these experiments.

---

### Decision · Program_Chairs · 2023-09-21

**Decision:**

Accept (poster)

**Comment:**

While the paper has a negative score, in overall the reviewers and I think the contribution is significant enough. In particular, it points out a legitimate bug in a paper (http://proceedings.mlr.press/v23/kifer12/kifer12.pdf) written 11 years back. One point to mention is that the same idea of a fix for GLMs was presented in a COLT paper (https://proceedings.mlr.press/v195/agarwal23d.html) this year. However, the reviewers, and I  are willing to consider this paper to be independent, since the NeurIPS submission deadline was prior to COLT conference.